# Whole genome sequence analysis of blood lipid levels in >66,000 individuals

Blood lipids are heritable modifiable causal factors for coronary artery disease. Despite well-described monogenic and polygenic bases of dyslipidemia, limitations remain in discovery of lipid-associated alleles using whole genome sequencing (WGS), partly due to limited sample sizes, ancestral diversity, and interpretation of clinical significance. Among 66,329 ancestrally diverse (56% non-European) participants, we associate 428M variants from deep-coverage WGS with lipid levels; ~400M variants were not assessed in prior lipids genetic analyses. We find multiple lipid-related genes strongly associated with blood lipids through analysis of common and rare coding variants. We discover several associated rare non-coding variants, largely at Mendelian lipid genes. Notably, we observe rare *LDLR* intronic variants associated with markedly increased LDL-C, similar to rare *LDLR* exonic variants. In conclusion, we conducted a systematic whole genome scan for blood lipids expanding the alleles linked to lipids for multiple ancestries and characterize a clinically-relevant rare non-coding variant model for lipids.

The discovery of rare alleles linked to plasma lipids (i.e., low-density lipoprotein cholesterol [LDL-C], high-density lipoprotein cholesterol [HDL-C], total cholesterol [TC], and triglycerides [TG]) continue to yield important translational insights toward coronary artery disease (CAD), including *PCSK9* and *ANGPTL3* inhibitors now available in clinical practice[1–5]. The monogenic and polygenic bases of plasma lipids are well-suited to population-based discovery analyses and confer broader insights for genetic analyses of complex traits. We now evaluate numerous newly catalogued, largely rare, alleles never previously systematically analyzed with lipids.

Analyses of imputed array-derived genome-wide genotypes and whole exome sequences in hundreds of thousands of increasingly diverse individuals continue to uncover low-frequency protein-coding variants linked to lipids. Due to purifying selection, causal variants conferring large effects tend to occur relatively more recently, and are thus rare and often specific to families or communities[6]. Most discovery analyses for large-effect rare alleles have focused on the analysis of disruptive protein-coding variants given (1) well-recognized constraint in coding regions, (2) incomplete genotyping of rare non-coding sequence given relative sparsity of deep-coverage (i.e., >30X) whole genome sequencing (WGS), and (3) better prediction of coding versus non-coding sequence

variation consequence[1,7–12]. We recently described a statistical framework incorporating multi-dimensional reference datasets paired with genomic data to improve rare coding and non-coding variant analyses for WGS analysis of lipids and other complex traits[13,14]. Furthermore, including individuals of non-European ancestry facilitates the discovery of both novel alleles at established loci as well as novel loci[14–16].

Here, we examine the full allelic spectrum with plasma lipids using whole genome sequences and harmonized lipids from the National Heart, Lung, and Blood Institute (NHLBI) Trans-Omics for Precision Medicine (TOPMed) program[17,18]. We studied 66,329 participants and 428 million variants across multiple ancestry groups—44.48% European, 25.60% Black, 21.02% Hispanic, 7.11% Asian, and 1.78% Samoan. We identified robust allelic heterogeneity at known loci with several novel variants at these loci; we additionally identified novel loci and pursued replication in independent cohorts. We then explored the association of genome-wide rare variants with lipids, with detailed explorations of rare coding and non-coding variant models at known Mendelian dyslipidemia genes. Our systemic effort yields new insights for plasma lipids and provides a framework for population-based WGS analysis of complex traits.

✉ e-mail: gpeloso@bu.edu; pnatarajan@mgh.harvard.edu

## Results

### Overview

We studied the TOPMed Freeze8 dataset of 66,329 samples from 21 studies and performed genome-wide association studies (GWAS) separately for the four plasma lipid phenotypes (i.e., LDL-C, HDL-C, TC, and TG) using 28 M individual autosomal variants (minor allele count [MAC] >20) and aggregated rare autosomal variant (minor allele frequency [MAF] <1%) association testing for 417 M variants (Fig. 1, Supplementary Fig. 1). Secondarily, we associated individual variants with minor allele frequencies (MAF) >0.01% within each ancestry group to detect ancestry-specific lipid-associated alleles. We intersected our results with currently published array-based GWAS results[15] to identify novel associations with lipids. We performed replication analyses for the putative novel associations identified, in up to ~45,000 independent samples with array-based genotyping imputed to TOPMed and 400 K samples from UK Biobank (UKB) imputed genotypes. Finally, we conducted rare variant association studies as multiple aggregate tests across the genome to identify gene-specific functional categories and non-coding genomic regions influencing plasma lipid concentrations. We replicated the significant rare variant aggregates in ~130 K whole genomes from UKB.

### TOPMed baseline characteristics

The TOPMed Informatics Research Center (IRC) and TOPMed Data Coordinating Center (DCC) performed quality control, variant calling, and calculated the relatedness of population structures of Freeze 8 data[17]. We studied 66,329 samples across 21 cohorts, and 41,182 (62%) were female. The ancestry distribution was 29,502 (44.46%) White, 16,983 (25.60%) Black, 13,943 (21.02%) Hispanic, 4719 (7.11%) Asian, and 1182 (1.78%) Samoan (Supplementary Data 1). The mean (standard deviation [SD]) age of the full cohort was 53 (15.00) years which varied by cohort from 25 (3.56) years for Coronary Artery Risk Development in Young Adults (CARDIA) to 73 (5.38) years for Cardiovascular Health Study (CHS). The Amish cohort had a higher-than-average concentration of LDL-C (140 [SD 43] mg/dL) and HDL-C (56 [SD 16] mg/dL) as well as lower TG (median 63 [IQR 50] mg/dL) consistent with the known founder mutations in APOB and APOC3[7,8,14]. In the Women's Health Initiative (WHI) cohort, the TC (230 [SD 41] mg/dL) and TG (median 129 [IQR 87] mg/dL) concentrations were higher than for other cohorts as previously described[12]. We accounted for lipid-lowering medications and fasting status and inverse rank normalized the phenotypes as before[12,14] which are further detailed in the Methods. The adjusted normalized lipid concentrations for the four lipids were similar across the cohorts.

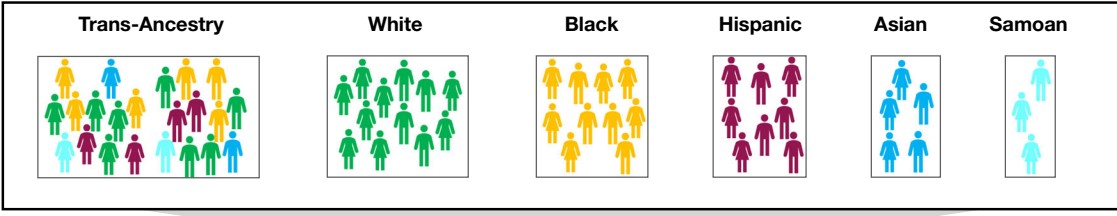

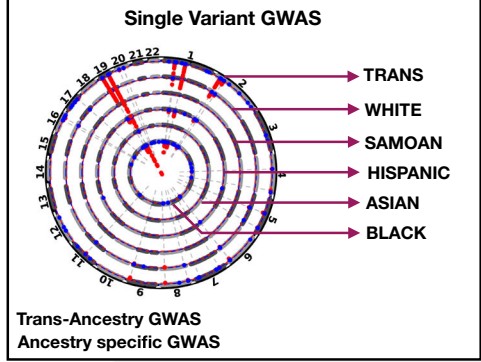

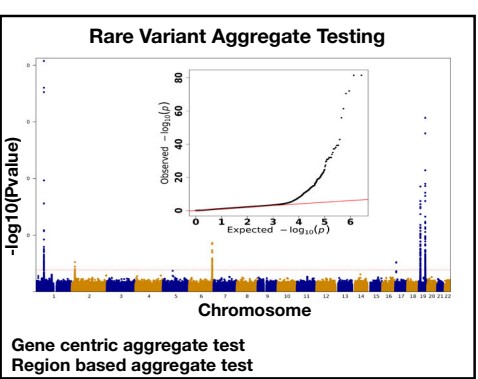

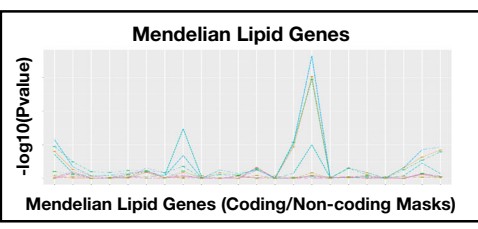

**Fig. 1 | Overall study schematic.** The analyses were conducted using the multi-ancestral TOPMed freeze8 data to associate whole genome sequence variation with lipid phenotypes (i.e., LDL-C, HDL-C, TC, and TG). A total of 66,329 samples with lipids quantified data from five ancestry groups were analyzed. Single variant GWAS were carried out using SAIGE on the Encore platform using SNPs with MAC >20. Both trans-ancestry and ancestry-specific GWAS were conducted. Genome-wide rare variant (MAF <1%) gene-centric and region-based aggregate tests were grouped and analyzed using STAARpipeline. Finally, single variant and rare variant associations at Mendelian dyslipidemia genes were investigated in further detail. TOPMed Trans-Omics for Precision Medicine, HDL-C high-density lipoprotein cholesterol, LDL-C low-density lipoprotein cholesterol, TC total cholesterol, TG triglycerides, GWAS genome wide association study, SAIGE Scalable and Accurate Implementation of GEneralized mixed model, MAC minor allele count, MAF minor allele frequency, SNPs single nucleotide polymorphisms.

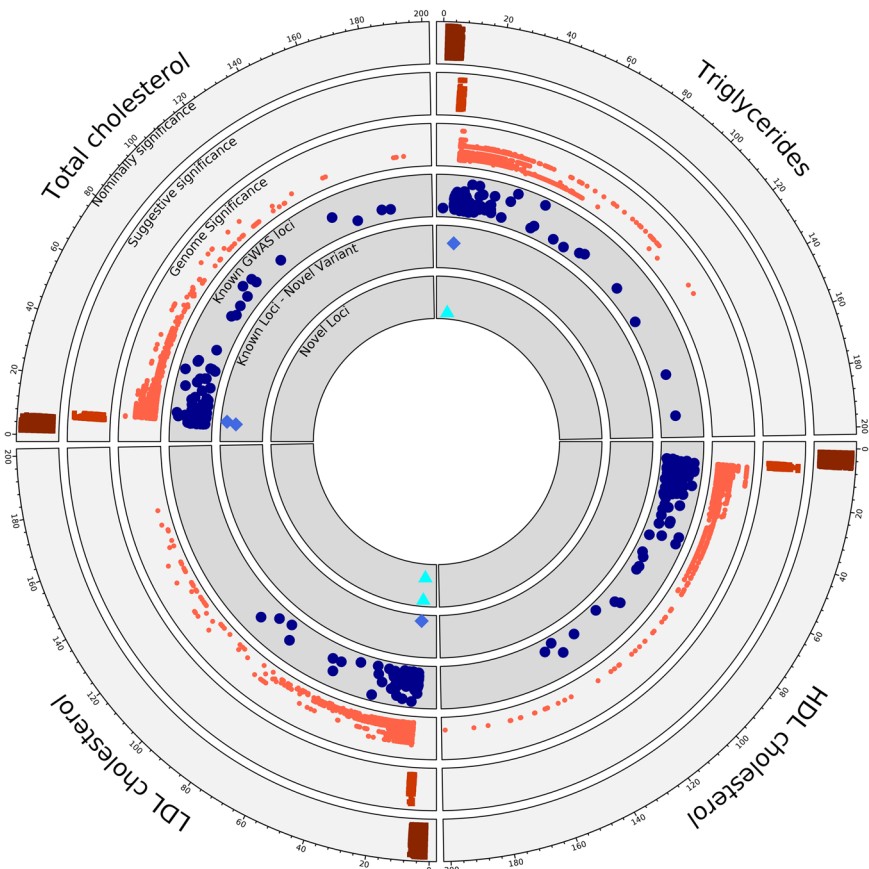

**Fig. 2 | Summary of single variant genome-wide association.** Representation of the single variant GWAS results from TOPMed Freeze 8 whole genome sequenced data of 66,329 samples. Each quarter represents a different lipid phenotype, and dots extending in clock-wise fashion represent variants with increasing evidence of association as noted by −log10(*p*-value), which was truncated at 200. The outer three circles show the GWAS data from TOPMed freeze8 where variants binned to nominally significant (*p*-value 0.05−5 × 10^{−07}), suggestive significant (*p*-value 5 × 10^{−07}−5 × 10^{−09}) and genome wide significant (*p*-value < 5 × 10^{−09}). The inner three circles compare our TOPMed results with known significantly associated lipid loci and variants from the MVP summary statistics and GWAS catalog to the identified novel variants and loci that are genome-wide significant from the current study, respectively. The figure represents the outputs from two-sided genetic association testing preformed using SAIGE-QT model, where the model was adjusted for all the covariates; see Methods. TOPMed Trans-Omics for Precision Medicine, GWAS genome wide association study, MVP million veteran program.

A total of 428 M variants passed the quality criteria with an average depth >30X in 22 autosomes. 202 M variants were singletons, 417 M were rare variants (MAF <1%), and 11 M were common or low-frequency variants (MAF >1%) with differences by cohort (Supplementary Data 2).

**Individual variant associations with lipids**

We performed single variant analysis of ~28 M variants with a MAC > 20 for four lipid phenotypes. We identified significant genomic risk loci for each lipid level (Supplementary Data 3) and considered a p-value <5 × 10^{−9} to claim significance as previously recommended for whole genome sequencing common variant association studies[14,19]. The total numbers of variants that met our significance threshold were 2214, 2314, 2697, and 2442 for LDL-C, HDL-C, TC and TG, respectively, and after clumping[20] the numbers of variants were 357, 338, 324, and 289, respectively. Of these variants, 99% were previously demonstrated to be associated with plasma lipids either at the variant- or locus-level[15] (Supplementary Data 4, Supplementary Fig. 2).

To identify putative novel variant associations, we compared our results to a recent multi-ethnic lipid GWAS among 312,571 participants of the Million Veteran Program (MVP)[15] as well as the GWAS Catalog (All associations(v1.0) file dated 06/04/2020) (Fig. 2). We clumped (window 250 kb, *r²* 0.5) significant variants using Plink[20] and queried these in the GWAS Catalog and MVP. Among genome-wide significant

variants, we tabulated 'known-position' (variant previously associated), 'known-loci' (variants not previously significantly associated with the corresponding lipid phenotype but within 500 kb of a known locus, thereby representing additional allelic heterogeneity), and 'novel' variants (variants not in a known lipid locus) (Supplementary Data 4).

The novel variants, tabulated in Table 1, are divided into two subsets—'novel variants' or variants at established lipid loci for another lipid phenotype, and 'novel loci,' representing new loci associations for any lipid phenotype. For example, the *CETP* locus is well-known for its link to HDL-C, but we now found that rs183130 (16:56957451:C:T, MAF 28.3%) at the locus is associated with LDL-C. Similarly, the variants rs7140110 (13:113841051:T:C, MAF 27.8%) *GAS6* and rs73729083 (7:137875053:T:C, MAF 4.5%) *CREB3L2* are newly associated with TC, while previous studies showed that rs73729083 associates with LDL-C[21] and rs7140110 associates with LDL-C[22] and TG[23]. Index variants at novel loci were typically low-frequency variants often observed in non-European ancestries, so we also conducted ancestry-specific association analyses for these alleles (Supplementary Data 5). For example, 12q23.1 (12:97352354:T:C, MAF 0.3%) and 4q34.2 (4:176382171:C:T, MAF 0.2%) associations with LDL-C are specific to Hispanic (MAF 1.3%) and Black (MAF 0.6%) populations, respectively and among Asians (MAF 1.5%) alone, 11q13.3 (11:69219641:C:T, MAF 0.2%) was associated with TG. One variant initially passing the novel locus filter for HDL-C (*RNF111*

**Table 1 | Putative novel variants identified in TOPMed and evidence for replication**

| Associated lipid phenotype | Novel variant class | Variants (Gene) | Discovery Cohort TOPMed Freeze8 (N = 66,329) | | | Replication Cohort Meta Analysis (METASOFT) MGB Biobank (N = 25,137); Penn Medicine Biobank (N = 20,079); UK Biobank (N = 424,955) | | |
|---|---|---|---|---|---|---|---|---|
| | | | Effect estimate | p-value | MAF | Beta | p-value | Std.Err |
| LDL-C | Novel locus | 12:97352354:T:C | −12.439 | $4.88 \times 10^{-09}$ | 0.003 | 3.316 | $3.62 \times 10^{-01}$ | 3.634 |
| LDL-C | Novel variant | 16:56957451:C:T (CETP) | −1.568 | $2.88 \times 10^{-09}$ | 0.283 | −1.459 | $8.74 \times 10^{-84}$ | 0.075 |
| LDL-C | Novel locus | 4:176382171:C:T | −16.086 | $2.82 \times 10^{-09}$ | 0.002 | −0.980 | $7.80 \times 10^{-01}$ | 3.514 |
| TC | Novel variant | 13:113841051:T:C (GAS6) | 1.731 | $1.12 \times 10^{-09}$ | 0.278 | 1.262 | $1.29 \times 10^{-38}$ | 0.097 |
| TC | Novel variant | 7:137875053:T:C (CREB3L2) | −4.106 | $7.54 \times 10^{-11}$ | 0.045 | −3.538 | $7.70 \times 10^{-07}$ | 0.716 |
| TG | Novel locus | 11:69219641:C:T | 0.232 | $1.98 \times 10^{-09}$ | 0.002 | −0.030 | $6.04 \times 10^{-01}$ | 0.059 |
| TG | Novel variant | 13:107551611:C:T (FAM155A) | 0.052 | $6.78 \times 10^{-10}$ | 0.045 | 0.015 | $2.20 \times 10^{-02}$ | 0.006 |

Variants identified as novel after comparing with the GWAS catalog and MVP summary statistics for associations with lipid phenotypes, including LDL-C, TC, and TG. All effect estimates are in mg/dL units, except for TG which was log-transformed in analysis thereby representing fractional change. Variants are categorized as novel loci or novel variant (i.e., known locus associated with another lipid phenotype) and the genes assigned to the variants per TOPMed whole genome sequence annotations (WGSA) are listed. Data is provided for the discovery (TOPMed freeze8) and replication cohorts (Imputed datasets from MGB Biobank, Penn Medicine Biobank and UK Biobank). Meta-analysis with the replication cohorts was carried out and the corresponding beta, p-values and standard-errors are provided. All the effect-estimates and p-values are reported from two-sided association testing with all independent samples from each cohort (Discovery-TOPMed: 66,329; Replication-MGB Biobank: 25,137; UK Biobank: 424,955; Penn Biobank: 20,079).
GWAS genome wide association study, MVP million veteran program, LDL-C low-density lipoprotein cholesterol, TC total cholesterol, TG triglycerides, TOPMed trans-omics for precision medicine, WGSA whole genome sequence annotations.

- rs112147665, beta = 8.664, p-value = $6.51 \times 10^{-10}$, was in LD (r = 0.7) with LIPC p.Thr405Met (rs113298164) which is known to be associated with HDL-C. The lead variant from MVP was 604 kb away from the *RNF111* variant but the rare *LIPC* missense variant p.Thr405Met was 421 kb away. Conditional analysis accounting for *LIPC* p.Thr405Met rendered the non-coding variant near *RNF111* variant non-significant (beta = 4.351, p-value = $2.47 \times 10^{-02}$), therefore we reclassified *RNF111* variant as a known-position variant. Ancestry-specific GWAS did not yield additional novel loci beyond our larger trans-ancestry GWAS. The majority of genome-significant single variants were captured by previous lipid GWAS[15], but ancestry-specific novel-hits are unique to WGS TOPMed data.

For the single variant GWAS, we pursued replication with two genome-wide array-based genotyped datasets imputed to TOPMed WGS[17,24]: Mass General Brigham (MGB) Biobank (N = 25,137) and Penn Medicine Biobank (N = 20,079)[25,26], these replication cohorts had diverse ancestry distribution, where non-European samples accounted for 15.77% in MGB Biobank and 51.20% in Penn Medicine Biobank. We also conducted replication using UKB imputed data which accounted for 16.10% of non-European samples (Supplementary Data 6). We brought seven putative novel variants with p-values < $5 \times 10^{-9}$ forward for replication. The three common variants, rs183130 (*CETP*), rs7140110 (*GAS6*), and rs73729083 (*CREB3L2*), that were associated with both LDL-C and TC in TOPMed replicated in MGB and UKB along with rs77687061 for TG and two of these (rs183130, rs73729083) replicated in Penn Biobank at an alpha level of 0.05 and consistent direction of effect (Supplementary Data 5). The two variants that were associated in all three replication studies were most significantly associated among African Americans in TOPMed (rs183130: beta = −2.762 mg/dL, p-value = $5.71 \times 10^{-07}$; rs73729083: beta = −3.725 mg/dL, p-value = $5.25 \times 10^{-07}$). We meta-analyzed the single variant replication from the three cohorts and identified three common variants with suggestive p-value ($5 \times 10^{-5}$) (Table 1). Low-frequency variants from specific ancestry groups associated with lipids in TOPMed were not replicated but we cannot rule out the possibility of reduced power due to the general underrepresentation of non-white ancestry groups in the replication data. In exploratory analyses, we extended the same approach for variants discovered to have $5 \times 10^{-9} < p\text{-value} < 5 \times 10^{-7}$ but did not observe replication (Supplementary Data 7).

## In-silico analysis to gain mechanic insights from single variant GWAS results

**Prioritization and functional enrichment analysis.** We first mapped the variants to genes and to functional regions using ANNOVAR. Second, we determined gene tissue specificity, relating tissue-specific gene expression with disease-gene associations, using MAGMA. Significantly associated variants were enriched in intronic and intergenic regions (Supplementary Fig. 3). Using GTEx, tissue-specific gene expression was enriched among liver, stomach, and pancreatic tissues (Supplementary Fig. 4) with top tissue-gene sets tabulated in Supplementary Data 8. Using the STRING protein-protein interaction database examining liver-specific genes, we highlight that the HDL-C protein network uniquely harbored metal-ions related genes (*MT1A, MT1B, MF1F, MT1G, MT1H*) and anticipated *LCAT-CETP* interactions (Supplementary Fig. 5). Enriched pathways from Reactome, GeneOntology and other curated and canonical pathways (Supplementary Data 9) with a p-value < $2.5 \times 10^{-06}$ were observed including response to metal ions, lipoprotein assembly, and chylomicron remodeling.

The enrichment analysis was carried out with the full single variant summary statistics, where we identified that most of the prioritized loci/genes were previously documented for lipid associations. Next, we specifically investigated the novel variants that we identified from this study. Out of the seven variants documented in Table 1, four were low frequency variants, 12:97352354:T:C (rs189010847) closest to *NEDD1*, 4:176382171:C:T (rs115489644) closest to *SPCS3*, 11:69219641:C:T (rs74791751) near to *SMIM38*, are all intergenic variants and 13:107551611:C:T (rs77687061) is an intronic variant in *FAM155A*. We did not find any information for these variants in the Open Target Genetics database[27]. Finally, two of the common novel-loci variants (rs183130 and rs7140110) were present in eQTL and sQTL databases[28], therefore, we performed analysis to determine the correlation among effects and the importance of these variants more in detail.

***CETP* locus, HDL-C, and LDL-C.** *CETP* is a well-recognized Mendelian HDL-C gene and the locus was previously known to be significantly associated with HDL-C, TC, and TG at genome-wide significance[15]. Pharmacologic *CETP* inhibitors have shown strong associations with increased HDL-C but mixed effects for LDL-C reduction in clinical trials[29–32]. We found that the *CETP* locus variant rs183130 (chr16:56957451:C:T, MAF 28.3%, intergenic variant) was associated

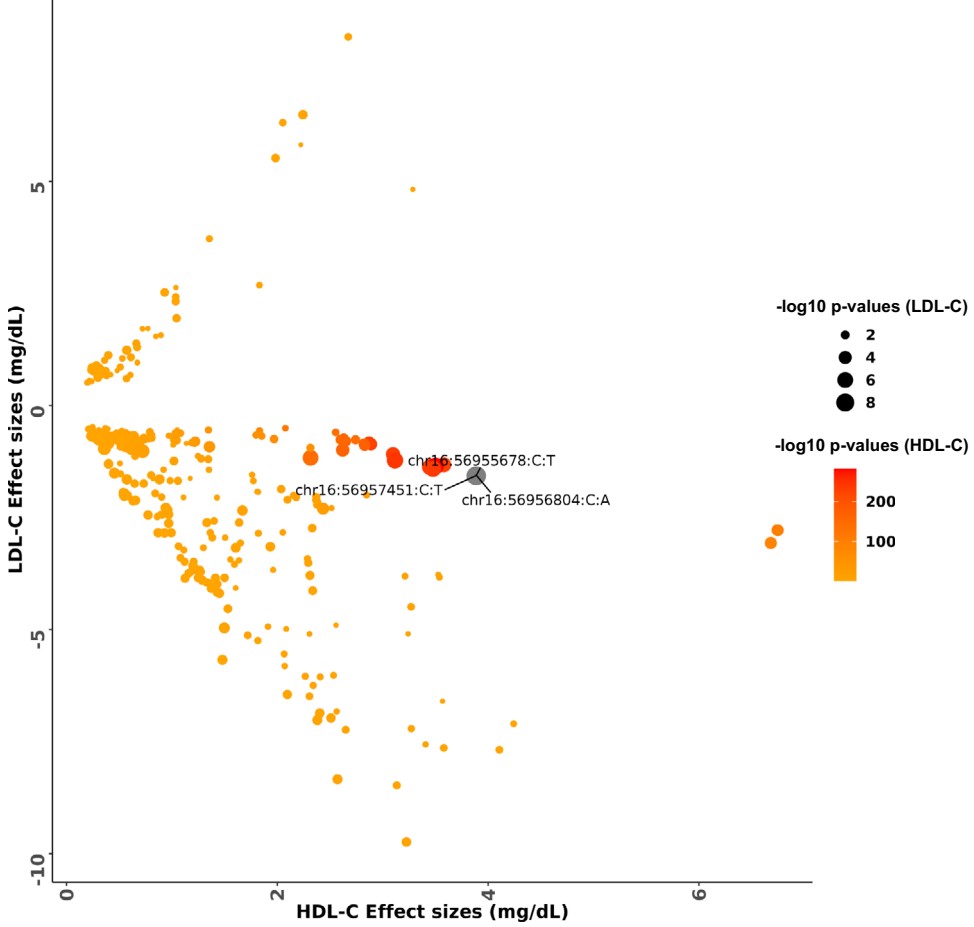

**Fig. 3 | Comparison of effects estimates for HDL-C and LDL-C among variants in the *CETP* locus.** The color scale of the data points was based on −log10 *p*-values from HDL-C association and the size of each data point was based on −log10 *p*-values of LDL-C association. Variants which are genome wide significant with LDL-C are represented as chromosome:position:reference allele:alternate allele. The effect estimates and p-values were calculated from two-sided genetic association testing preformed using SAIGE-QT model, where the model was adjusted for all the covariates; see Methods. HDL-C high-density lipoprotein cholesterol, LDL-C low-density lipoprotein cholesterol.

with reduced LDL-C concentration (beta = −1.568 mg/dL, SE = 0.264, *p*-value = 2.88 × 10⁻⁰⁹). The lead HDL-C-associated variant at the locus, rs3764261 (chr16:56959412:C:A, MAF 30.3%), was associated with 3.5 mg/dL increased HDL-C (*p*-value = 8.03 × 10⁻²⁸³), and rs183130 was associated with 3.9 mg/dL increased HDL-C (*p*-value < 1 × 10⁻²⁸⁴) as well. Among the ancestry groups analyzed, rs183130 was most significantly associated with LDL-C among those of African ancestry (beta = −2.762 mg/dL, *p*-value = 5.71 × 10⁻⁰⁷) (Supplementary Data 10). We next investigated variants by their HDL-C and LDL-C effects within this locus (+/−500 kb of rs183130 and rs3764261) (Fig. 3). We identified five variants showing at least suggestive (*p*-value < 5 × 10⁻⁰⁷) association with both HDL-C and LDL-C. Though variants with strong LD (linkage disequilibrium) existed, ancestry-specific analyses showed that the stronger LDL-C effects were among those of African ancestry.

To better understand the mechanisms for HDL-C and LDL-C effects at the *CETP* locus, we pursued colocalization with eQTLs from three tissues (Liver, Adipose Subcutaneous and Adipose Visceral [Omentum]) from GTEx[28]. We analyzed 5 LDL-C and 441 HDL-C associated (*p*-values <5 × 10⁻⁰⁷) variants. We correlated eQTL effect estimates for genes at the locus with lipid outcome effect estimates. Indeed, *CETP* gene expression effects were strongly negatively correlated with HDL-C effects (Liver: ρ −0.933, *p*-value 4.01 × 10⁻¹⁷; Adipose Subcutaneous: ρ −0.762, *p*-value 8.87 × 10⁻¹²; Adipose Visceral: ρ −0.739, *p*-value 5.52 × 10⁻¹⁰) (Supplementary Fig. 6). However, *CETP* expression effects were not significantly correlated with LDL-C (Liver: ρ 0.007, *p*-value 0.99; Adipose

Subcutaneous: ρ 0.344, *p*-value 0.57; Adipose Visceral: ρ −0.59, *p*-value 0.29). Given the possibility that the observed lack of correlation for LDL-C could be due to reduced power from a limited number of variants attaining a suggestive *p*-value (<5 × 10⁻⁰⁷), we repeated the analysis with a subset of 122 nominally significant (*p*-value < 0.05) LDL-C associated variants in this locus. Indeed, *CETP* gene expression effects were strongly positively correlated with LDL-C effects (Liver: ρ 0.957, *p*-value 2.28 × 10⁻⁰⁸; Adipose Subcutaneous: ρ 0.922, *p*-value 1.34 × 10⁻¹⁵; Adipose Visceral: ρ 0.868, *p*-value 6.09 × 10⁻¹¹).

**GAS6 locus, LDL-C/TG, and TC.** Variants at *GAS6* were previously associated with LDL-C and TG[22,23], but in our analysis, rs7140110 was now significantly associated with TC. We performed colocalization analysis of the variants+/−500 Kb from rs7140110 in liver and adipose tissues from GTEx. Across the three lipid-related tissues (liver, adipose subcutaneous, and adipose visceral), strong colocalization was observed in liver for all three lipid phenotypes (TG 46.6%; LDL-C 33.3%; TC 28%). The TG and LDL-C-associated variants were eQTLs for the *GAS6* gene only. However, the TC-associated eQTLs at this locus influenced the *cis* expression of multiple genes, including *GAS6*, antisense genes of *GAS6* (AS1, AS2) as well as other genes (i.e., *TFDP1, CHAMP1, LINC00565, ADPRHL1, RASA3, UPF3A, GRTP1, AL442125.1, C13orf46, DCUN1D2, CDC16, TMEM255B, GRTP1-AS1, ATP4B, TMCO3*). In addition to *GAS6*, the TC-associated rs7140110 is an sQTL for *TMEM255B* in adipose subcutaneous tissue (*p*-value 5.6 × 10⁻⁰⁸), with

further support from TC colocalization analysis and was not significant for other lipid levels.

**Phenome-wide association with complex traits.** We conducted a phenome-wide association (PheWAS) of 1572 binary complex traits using UK Biobank for the three replicated common variants (16:56957451:C:T (*CETP*); 13:113841051:T:C (*GAS6*); 7:137875053:T:C (*CREB3L2*)) adjusting for PC1–10, age, age$^2$, sex, and race. We claimed significance at FDR of 0.05 and identified various complex traits significant, including ischemic heart disease for the *CETP* variant and heart failure/atherosclerosis, hypercholesterolemia traits for *GAS6* variant. The summary statistics from PheWAS analysis for the significant complex traits are tabulated in Supplementary Data 11.

### Rare variant aggregates associated with lipids

**Gene-Centric associations.** We next evaluated the association of aggregated rare (MAF < 1%) variants, linked to protein-coding genes ('gene-centric'). We employed a Bonferroni-corrected significance threshold of $0.05/20{,}000 = 2.5 \times 10^{-06}$ for coding and non-coding gene-centric rare variant analyses (Supplementary Fig. 7). We identified 102 coding and 160 non-coding gene-centric rare variant aggregates significantly associated with at least one of the four plasma lipid phenotypes in nonconditional analysis (Supplementary Data 12, 13). We secondarily conditioned our significant aggregate sets on variants individually associated with lipid levels from the GWAS catalog, MVP summary statistics and the TOPMed data. We identified 74 coding and 25 non-coding rare variants aggregates associated with at least one lipid level after conditional analyses (Supplementary Data 14, 15).

Most of the coding gene-centric sets remained significant after secondary conditioning, while a minority of non-coding gene-centric sets remained significant after conditioning. Significant genes identified from coding rare variant analyses included multiple known Mendelian lipid genes including *LCAT*, *LDLR*, and *APOB* (Supplementary Data 13). *RFC2* putative loss-of-function mutations (combined allele frequency < 0.002%) were significantly associated with triglycerides (*p*-value $2 \times 10^{-06}$) representing a putative novel association for triglycerides. The *RFC2* aggregate set (plof) was associated with reduced TG (beta = −0.89 for log[TG]). The persistently significant regions identified from non-coding rare variant analyses linked to genes included the UTR (untranslated region) for *CETP* and promoter-CAGE (CAGE−Cap Analysis of Gene Expression sites) around *APOA1* for HDL-C, and *APOE* promoter-CAGE, *APOE* enhancer-DHS (DHS−DNase hypersensitivity sites), and *EHD3* promoter-DHS for total cholesterol (Supplementary Data 15). Most of the coding aggregates had larger effects compared to non-coding aggregates, and among the non-coding aggregates SPC*24* non-coding aggregate (enhancer-CAGE) at the *LDLR* locus had the strongest effect for LDL-C (beta = 2.320 mg/dL; *p*-value = $1.75 \times 10^{-05}$).

We analyzed the UK Biobank whole genome sequences among ~130 K participants to provide evidence of replication for the significant coding and non-coding aggregate sets. We used a Bonferroni-corrected significance threshold based on the number of genes tested in each type of aggregate-based test. For gene centric-coding aggregates, we conducted replication of 21 genes (*p*-value < 0.05/ $21 = 2.38 \times 10^{-03}$) and for non-coding aggregates we replicated the findings from 13 genes (*p*-value < 0.05/13 = $3.85 \times 10^{-03}$). At Bonferroni significance, 71% and 62% of genes replicated for at least one coding and non-coding aggregate set, respectively (Supplementary Data 14, 15). We observed that most of the Mendelian lipid genes replicated for coding aggregates including *ABCA1, ABCG5, LCAT, APOB, LDLR, PCSK9,* and *LPL*. For the non-coding aggregate set, the most significant replications were observed for the *APOB*, *LDLR* (SPC*24*), and *PCSK9* loci, further corroborating the observation that both coding and noncoding rare variant signals contribute to variation in lipid levels at these loci.

**Region-based associations.** We also performed unbiased region-based rare variant association analyses tiled across the genome with both static and dynamic window sizes. We first evaluated 2.6 M regions statically at 2 kb size and 1 kb window overlap by the sliding window approach. Statistical significance was assigned at $0.05/(2.6 \times 1^{-06})$ = $1.88 \times 10^{-08}$. We identified 28 significantly associated windows with at least one lipid phenotype. After conditioning on variants individually associated with the corresponding lipid phenotype, we identified two regions at *LDLR* still significantly associated with both total cholesterol and LDL-C, although these regions included both intronic and exonic variants (Supplementary Data 16). *LDLr* intron 1, which encodes *LDLR-AS1* (LDLR antisense RNA 1) on the minus strand, had suggestive evidence for association with TC (*p*-value $3.17 \times 10^{-6}$) with −2.76 mg/dL reduction in TC. A prior study identified that a common variant (rs6511720, MAF 0.11) in *LDLR* intron 1 is associated with increased *LDLR* expression in a luciferase assay and reduction in LDL-C[33]. When adjusting for rs6511720, the significance improved (*p*-value $1.43 \times 10^{-8}$) with −3.35 mg/dL reduction in TC.

For dynamic window scanning of the genome, we implemented the SCANG method[34]. The SCANG procedure accounts for multiple testing by controlling the genome-wide error rate (GWER) at 0.1[34]. In the dynamic window-based workflow, STAAR-O detected 51 regions significantly associated with at least one lipid phenotype after conditioning on known variants (Supplementary Data 17). Most of the regions mapped to known Mendelian lipid genes, including *LCAT* ($8.7 \times 10^{-13}$) for HDL-C, and *LDLR* ($2.4 \times 10^{-28}$, $7.3 \times 10^{-26}$) and *PCSK9* ($2.9 \times 10^{-12}$, $5.5 \times 10^{-12}$) for LDL-C and TC, respectively. Exon 4 aggregates of *LDLR* were specifically associated with 20 mg/dL increase in LDL-C. *PCSK9* Exon2-Intron2 region spanning chr1:55043782–55045960 had significantly reduced LDL-C by 6 mg/ dL (*p*-value = $3 \times 10^{-13}$), and the effect persisted even with only Intron 2 rare variants of *PCSK9* (−5 mg/dL, *p*-value = $2 \times 10^{-8}$). Strikingly, in secondary analyses, we found evidence for very large effects for rare variants in *LDLR* Introns 2 and 3 (+21 mg/dL, *p*-value = $7 \times 10^{-4}$) and *LDLR* Introns 16 and 17 (+17 mg/dL, *p*-value = 0.02), similar to rare coding *LDLR* mutations. While 32 of the significant dynamic windows also included exonic regions, there were also several dynamic windows significantly independently associated with lipids not containing exonic regions. For example, four non-coding windows (two overlapping) at 2p24.1, which harbors the Mendelian *APOB* gene, were significantly associated with LDL-C. Intronic non-coding regions were associated with both LDL-C and TC -associated windows at *LPAL2-LPA-SLC22A3*; for example, *LPAL2* Intron 3 was associated with a 3.7 mg/dL increase in TC. Non-coding TC-associated significant dynamic windows were near *TOMM40/APOE*. One rare variant signal observed was at *TOMM40* Intron 6, where the 'poly-T' variant in this region is on the *APOE4* haplotype and influences expressivity for Alzheimer's disease age-of-onset[35,36]. For HDL-C, we identified significant non-coding windows at an intergenic region near *LPL* and *CD36* Intron 4. In the generation of the spontaneously hypertensive rat model, the deletion of intron 4 in *CD36* with resultant *CD36* deficiency has been mapped to defective fatty acid metabolism in this model[37]. Several regions significant in SCANG were not even nominally significant in burden association analyses indicating the likelihood of causal variants with bidirectional effects.

We replicated 28 sliding and 51 dynamic window aggregate sets using UKB whole genomes, at a Bonferroni-corrected alpha threshold of 0.05/no. of regions for each approach separately. At Bonferroni significance, 61% of the regions from each of the sliding window (*p*-value < 0.05/28 = $1.79 \times 10^{-03}$) and dynamic window (*p*-value < 0.05/ 51 = $9.80 \times 10^{-04}$) approaches significantly replicated (Supplementary Data 16, 17). Multiple regions linked to *LDLR, PCSK9, CETP, APOC3,* and *ABCA1* were highly significant.

Several gene-centric non-coding aggregates associated with lipids near known monogenic lipid genes but mapped to another gene at the

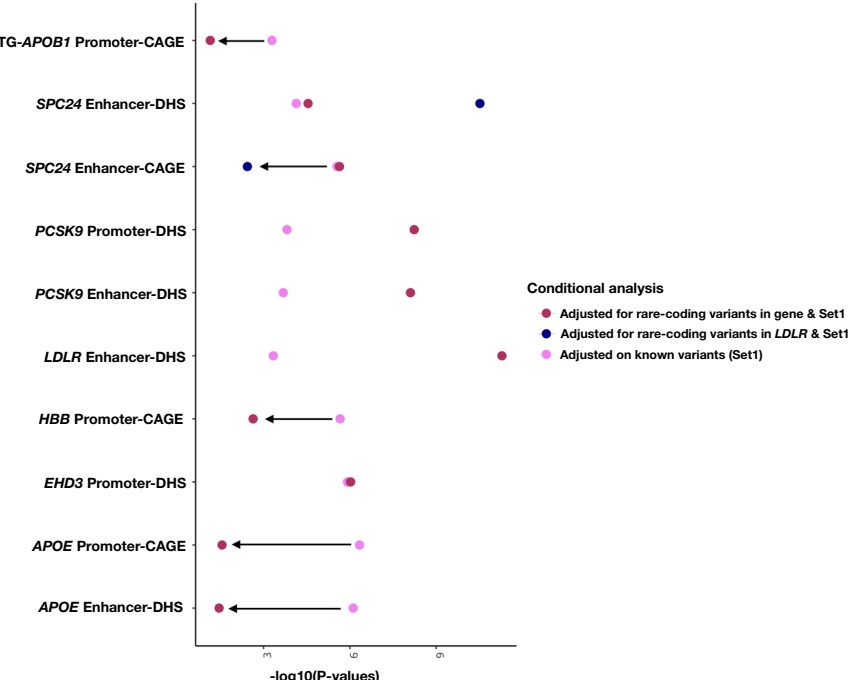

**Fig. 4 | Conditional analysis of coding rare-variants from the same gene and a near-by gene.** Non-coding rare variant sets significantly associated with TC and TG after the conditional analysis on known variants are shown with additional adjustment on rare-coding variants. The additional adjustment for rare-coding variants were carried out for the same gene of the aggregate set and for certain gene aggregates (SPC24) the conditional analysis was carried out with a nearby Mendelian gene. After adjusting for rare-coding variants and known variants, EHD3 signal drops minimally, whereas signal from PCSK9 (promoter-DHS, enhancer-DHS), LDLR-loci (enhancer-DHS, SPC24 enhancer-DHS) enhances significantly.

APOB1, SPC24 (enhancer-CAGE), HBB and APOE signal drops after the conditional analysis on rare-coding variants. The different colored dots on the plot represents the conditional STAAR-O p-values when adjusting for known variants (Set1) and rare-coding variants of the same or near-by gene. The p-values were calculated from two-sided aggregate testing preformed using STAAR gene-centric model, where the model was adjusted for all the covariates; see Methods. STAAR variant-Set Test for Association using annotation information, TC total cholesterol, TG triglycerides, CAGE cap analysis of gene expression, DHS DNase hypersensitivity.

locus via annotations. Therefore, we performed downstream conditional analyses adjusting the gene-centric non-coding results for rare coding variants (MAF < 1%) within known lipid monogenic genes (Supplementary Data 18). When accounting for both common and rare coding variants at the nearby familial hypercholesterolemia LDLR gene, SPC24-enhancer DHS was significantly associated with total cholesterol (p-value = $3.01 \times 10^{-11}$) and with suggestive evidence for LDL-C (p-value = $1.57 \times 10^{-06}$). In a similarly adjusted model, LDLR-enhancer-DHS showed a strong association with TC (p-value $5.18 \times 10^{-12}$). When adjusting for known common variants as well as rare coding variants in PCSK9, both PCSK9-enhancer DHS and PCSK9-promoter DHS were significantly associated with total cholesterol (Fig. 4, Supplementary Fig. 8). Through this procedure, CETP UTR retained significance for its independent association with HDL-C as well as the putatively novel gene EHD3-promoter DHS association with TC. However, the non-coding gene-centric APOC3 and APOE associations were rendered non-significant for HDL-C and TC, respectively.

Since we cannot rule out the possibility of reduced power for genome-wide rare variant analyses, we leveraged current knowledge of 22 Mendelian lipid genes for more focused exploratory analyses[14]. We validated most genes in rare variant coding analyses. The genes with the strongest coding signals typically had at least nominal evidence of gene-centric non-coding rare variant associations (Supplementary Data 19, Supplementary Fig. 9). When rare coding variants were introduced into the model, the evidence for non-coding rare variant associations were largely unchanged. Our findings expanding the currently described genetic basis for hypercholesterolemia to include rare non-coding variation at LDLR and PCSK9 (Fig. 5).

### Heritability contributions from rare variants

To understand the contribution of rare variants towards lipid trait heritability, we examined heritability of lipids by variant allele frequency across three ancestral samples (White, Black, and Hispanic) in TOPMed. We calculated trait heritability using Greml-LDMS[38] following the steps as implemented by Wainschtein et al.[39]. Using the TOPMed WGS, we grouped the variants into 4 MAF bins for the three ancestral samples. In each MAF bin, we grouped variants based on the LD scores into four quartiles and calculated variance contributed by the SNPs ($h^2$) for each of the lipids using unrelated individuals from each ancestral group (Supplementary Fig. 10) and set negative estimate to zero. We observed that rare variants from the lower MAF bins contributed to trait heritability but have large standard errors (Supplementary Data 20). We observed an increase in $h^2$ values including WGS variants relative to estimates obtained from array-genotypes as reported by Cadby et al.[40] for the European samples. We also compared the $h^2$ estimates from all the variants from WGS TOPMed cohort against array-genotypes captured in MGB Biobank to understand the differences contributed by these two sequencing methods. As expected, the $h^2$ estimates from array-genotypes were reduced corresponding to missing heritability from the lower MAF bins captured by WGS. The heritability estimates from array-genotypes were markedly higher for European samples relative to African and Hispanic sample sets indicating that WGS better captured heritability for the latter groups.

### Discussion

Conducting one of the largest population-based WGS association analyses, we now simultaneously interrogate and establish a common, rare coding, and rare non-coding variant model for a complex trait. Utilizing 66,329 diverse individuals with deep-coverage WGS, we

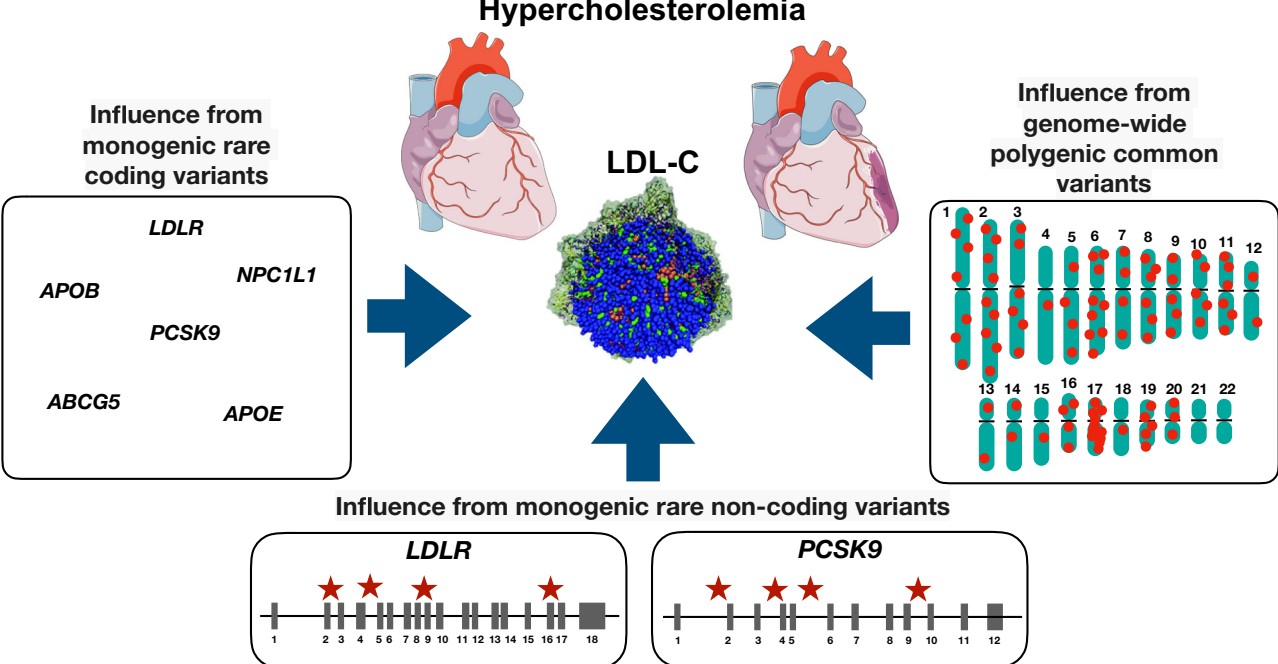

**Fig. 5 | Influence of common and rare variants with hypercholesterolemia.** In addition to monogenic contributions from rare variants in Mendelian hypercholesterolemia genes, multiple genome-wide significant LDL-C-associated common variants also yield a polygenic basis for hypercholesterolemia. In the present work, we now identify rare non-coding variants in proximity of Mendelian hypercholesterolemia genes, specifically *LDLR* and *PCSK9*, that also contribute to the genetic basis of hypercholesterolemia. Parts of the figure were generated using pictures from Servier Medical Art. Servier Medical Art by Servier is licensed under a Creative Commons Attribution 3.0 Unported License (https://creativecommons.org/licenses/by/3.0/). LDL-C low-density lipoprotein cholesterol.

interrogated 428 M variants with plasma lipids expanding the allelic series to rare non-coding variants, often within introns, of Mendelian lipid genes with prior robust rare coding variant support. Our observations have important implications for plasma lipids as well as the genetic basis of complex traits more broadly.

WGS of diverse ancestries enables both allelic and locus heterogeneity for complex traits. Population genetic analyses have largely been enriched for individuals of European descent[41]. Genetic association of plasma lipids using arrays or whole exome sequencing among Europeans have yielded several important insights regarding plasma lipids and the causal determinants of CAD[4,5,42–44]. Similar increasingly larger studies among non-Europeans have often yielded new genetic loci and sometimes new genes, such as *PCSK9*[1,15,16,45,46]. Such differences have also led to concerns about the use of polygenic risk scores gleaned from much larger European GWAS of complex traits for non-Europeans[47]. Aided by the availability of WGS data, we identify new putative loci associated with lipids in non-Europeans. Furthermore, our study enabled the discovery of several novel alleles at known loci, with richly distinct allelic heterogeneity across ancestry groups. For example, HDL-C-raising *CETP* locus variants linked to *CETP* gene expression were only associated with LDL-C reduction among those of African ancestry. While all pharmacologic *CETP* inhibitors increase HDL-C, only those that decrease LDL-C also reduce cardiovascular disease risk[29–32]. Given the contribution of genetic differences, clinical trials with more diverse samples would show insights.

Our study now provides increasingly robust evidence for a rare non-coding variant model for complex traits. Our rare non-coding variant associations in both gene-centric and sliding window models were largely restricted to the introns of Mendelian lipid genes with prior robust rare coding variant support consistent with biologic plausibility[48]. Rare intronic variants, often impacting splicing, have been previously implicated in afflicted Mendelian families or small exceptional case series, often through candidate gene approaches[49–52]. We discovered one example of a rare non-coding signal without prior rare coding support—i.e., *EHD3* which also nominally replicated in the independent UKB WGS cohort. We obtained estimates of phenotypic effect using burden tests. For most regions, even nominal significance was not detected using burden testing indicating the likelihood of variants with bidirectional effects further complicating clinical interpretation. When burden signals were detected, observed effects were typically larger than common non-coding variants and less than rare coding variants, with the exception of *LDLR*, consistent with whole genome mutational constraint models[53–55].

The detection of independent rare non-coding variant signals has remained elusive largely due to limited sample sizes with requisite WGS and limitations in the interpretation of rare non-coding variation functional consequence. Previously, we used annotated functional non-coding sequence in 16,324 TOPMed participants, and found that rare non-coding gene regions associated with lipid levels, but they were not independent of individually associated single variants[14]. Using STAAR, we observed putative rare non-coding variant associations for lipids independent of individual variants associated with lipids in TOPMed.

WGS can improve diagnostic yield beyond the current standard of next-generation gene panel sequencing for dyslipidemias. A very small fraction with severe hypercholesterolemia and features consistent with strong genetic predisposition have a familial hypercholesterolemia variant in *LDLR*, *APOB*, or *PCSK9*[56,57]. The presence of familial hypercholesterolemia variants is independently prognostic for CAD, beyond lipids, and merits the consideration of more costly lipid-lowering medications[56–59]. We now observe that rare *LDLR* variants in Introns 2, 3, 16, and 17 lead to ~0.5 standard deviation increase in LDL-C, approximating effects observed with clinically reported exonic familial hypercholesterolemia variants in *LDLR*[59]. Small studies have indicated the possibility of rare intronic *LDLR* variants causing familial hypercholesterolemia due to altered splicing, which we now observe in our unbiased population-based WGS study[60,61]. A WGS approach to lipid disorders, particularly for familial

hypercholesterolemia, will markedly improve the diagnostic yield beyond existing limited approaches.

Our dynamic window approach may also improve the clinical curation of exonic variants. Among the data used to curate exonic variants is the use of in silico functional prediction tools[62]. Although evolutionary constraint measures are typically employed, such tools are largely agnostic to functional domain. As it relates to lipids, disruptive *APOB* and *PCSK9* exonic variants can lead to strikingly opposing directions with large effects for LDL-C depending on locations[1,8,63,64]. Using SCANG[34], we detect a significant association with large effect for *LDLR* Exon 4 itself. This observation supports the pathogenicity of *LDLR* Exon 4 disruptive variants among patients with severe hypercholesterolemia. The majority of familial hypercholesterolemia variants worldwide occur in Exon 4 of *LDLR*[65–68]. Conventional rare coding variant analyses aggregate all exonic variants for a transcript. Here, we demonstrate an opportunity for exon-level rare variant association testing.

Our discovery analyses with replication as well as heritability assessment are consistent with the notion that both rare coding and non-coding alleles, not well-captured by genome-wide arrays. Furthermore, we observe that heritability gains relative to genome-wide genotyping arrays are more significant for individuals of European-ancestry likely indicative of Eurocentric array designs. A tradeoff for WGS, however, is the greater cost. However, as costs continue to decrease as well as cheaper WGS implementations via reduced coverage, cost may no longer be a downside.

Our study has important limitations. First, while our study is large for a WGS study by contemporary standards, it is dwarfed by existing GWAS datasets limiting power for novel discovery. Nevertheless, by using WGS in diverse ancestries, we can study hundreds of millions new variants. Second, prediction of rare non-coding variation consequence to prioritize causal variants remains a challenge thereby limiting power[69]. The striking difference for most STAAR and burden results also highlights bidirectional effects for rare non-coding variants within the same region and further challenges for clinical utility. Third, given the paucity of multi-ancestral WGS datasets with lipids, our analyses are largely restricted to TOPMed and replication to European rich UK Biobank WGS data. For single variant associations, we pursued TOPMed-imputed GWAS datasets but were limited by the lack of ancestral diversity. As TOPMed is a consortium of multiple different cohorts, we demonstrate consistencies by cohort. Furthermore, rare variant non-coding signals were largely restricted to regions with rare variant coding signals supporting biological plausibility.

In conclusion, using WGS and lipids among 66,329 ancestrally diverse individuals we expand the catalog of alleles associated with lipids, including allelic heterogeneity at known loci and locus heterogeneity by ancestry. We characterize the common, rare coding, and rare non-coding variant model for lipids and replicated the results. Lastly, we now demonstrate a monogenic-equivalent model for rare *LDLR* intronic variants predisposing to marked alterations in LDL-C, currently not recognized in current population or clinical models for LDL-C.

## Methods
### Dataset
**Contributing studies.** The discovery cohort includes the whole genome sequenced (WGS) data of 66,329 samples from 21 studies of the Trans-Omics for Precision Medicine (TOPMed) program with blood lipids available[17]. The overall goal of TOPMed is to generate and use trans-omics, including whole genome sequencing, of large numbers of individuals from diverse ancestral backgrounds with rich phenotypic data to gain novel insights into heart, lung, blood, and sleep disorders. The Freeze 8 data includes 140,306 samples out of which 66,329 samples qualified with lipid phenotype. Freeze 8 dataset passed the central quality control protocol implemented by the TOPMed

Informatics Research Core (described below) and was deposited in the dbGaP TOPMed Exchange Area.

The studies included in the current dataset, along with their abbreviations and sample sizes, contains the Old Order Amish (Amish, $n = 1083$), Atherosclerosis Risk in Communities study (ARIC, $n = 8016$), Mt Sinai BioMe Biobank (BioMe, $n = 9848$), Coronary Artery Risk Development in Young Adults (CARDIA, n = 3,056), Cleveland Family Study (CFS, $n = 579$), Cardiovascular Health Study (CHS, $n = 3,456$), Diabetes Heart Study (DHS, $n = 365$), Framingham Heart Study (FHS, $n = 3992$), Genetic Studies of Atherosclerosis Risk (GeneSTAR, $n = 1757$), Genetic Epidemiology Network of Arteriopathy (GENOA, $n = 1046$), Genetic Epidemiology Network of Salt Sensitivity (GenSalt, $n = 1772$), Genetics of Lipid-Lowering Drugs and Diet Network (GOLDN, $n = 926$), Hispanic Community Health Study - Study of Latinos (HCHS_SOL, $n = 7714$), Hypertension Genetic Epidemiology Network and Genetic Epidemiology Network of Arteriopathy (HyperGEN, $n = 1853$), Jackson Heart Study (JHS, $n = 2847$), Multi-Ethnic Study of Atherosclerosis (MESA, $n = 5290$), Massachusetts General Hospital Atrial Fibrillation Study (MGH_AF, $n = 683$), San Antonio Family Study (SAFS, $n = 619$), Samoan Adiposity Study (SAS, $n = 1182$), Taiwan Study of Hypertension using Rare Variants (THRV, $n = 1982$) and Women's Health Initiative (WHI, $n = 8263$) (Please see Supplementary Note 1 for additional details). The multi-ancestral data set included individuals from White (44%), Black (26%), Hispanic (21%), Asian (7%), and Samoan (2%) ancestries. Study participants granted consent per each study's Institutional Review Board (IRB) approved protocol. Secondarily, these data were analyzed through a protocol approved by the Massachusetts General Hospital IRB. Supplementary Data 1 details the number of samples across different studies and ancestral group.

The replication cohorts for single variant GWAS include TOPMed-imputed genome-wide array data from the Mass General Brigham (MGB), Penn Medicine Biobanks and UK Biobank (UKB) imputed data which consist of 25,137, 20,079, and 424,955 samples respectively[25,26,70]. The replication cohort for rare variant aggregates test include UKB whole genome sequenced data which consists of a subset of 133,360 UKB participants, where we removed unconsented and related individuals. We curated the MGB Biobank and Penn Medicine Biobank phenotype data from the corresponding electronic health record databases in accordance with corresponding institutional IRB approvals. The UKB data included volunteer residents of the UK aged 40–69 and were recruited between 2006 and 2010. Consent was previously obtained from each participant regarding storage of biological specimens, genetic sequencing, access to all available electronic health record (EHR) data, and permission to recontact for future studies. All UKB participants gave written informed consent per UKB primary protocol. The MGB Biobank consists of 54%, Penn Medicine Biobank consist of 52% and UK Biobank imputed data consist of 54% of female samples and average ages were 55.89, 58.35 and 56.55 years, respectively (Supplementary Data 6).

**Phenotypes.** The primary outcomes in this study included LDL cholesterol (LDL-C), HDL cholesterol (HDL-C), total cholesterol (TC), and triglycerides (TG) phenotypes. LDL-C was either directly measured or calculated by the Friedewald equation when triglycerides were <400 mg/dL. Given the average effect of lipid lowering-medicines, when lipid-lowering medicines were present, we adjusted the total cholesterol by dividing by 0.8 and LDL-C by dividing by 0.7, as previously done[14]. Triglycerides remained natural log transformed for analysis. Fasting status was accounted for with an indicator variable.

We harmonized the phenotypes across each cohort[18] and inverse rank normalization of the residuals of each race within each cohort scaled by the standard deviation of the trait and adjusted for covariates[12]. We included covariates such as age, age$^2$, sex, PC1–11, study-groups as well as Mendelian founder lipid variants *APOB*

p.R3527Q and *APOC3* p.R19X for the Amish cohort[7,8,71]. Supplementary Data 1 provides the distributions of each of the four lipid phenotypes by cohort, ancestral groups, and gender. For the UK Biobank, we curated the first instance of the four lipids (data field numbers: HDL-C-30760; LDL-C-30780; TC-30690; TG-30870). The lipid measurements from mmol/L were converted to mg/dL by multiplying TG measurements by 88.57 and for other lipids by multiplying by 38.67. We executed similar steps of phenotype harmonization and normalization for the replication cohorts. In addition, we adjusted the MGB Biobank for study-center and array-type, and Penn Medicine Biobank for ancestry and BMI in addition to the other common covariates.

**Genotypes.** Whole genome sequencing of goal >30X coverage was performed at seven centers (Broad Institute of MIT and Harvard, Northwest Genomics Center, New York Genome Center, Illumina Genomic Services, PSOMAGEN [formerly Macrogen], Baylor College of Medicine Human Genome Sequencing Center, and McDonnell Genome Institute [MGI] at Washington University). In most cases, all samples for a given study within a given Phase were sequenced at the same center (Supplementary Note 1). The reads were aligned to human genome build GRCh38 using a common pipeline across all centers (BWA-MEM).

The TOPMed Informatics Research Core at the University of Michigan performed joint genotype calling on all samples in Freeze 8. The variant calling "GotCloud" pipeline (https://github.com/statgen/topmed_variant_calling) is under continuous development and details on each step can be accessed through TOPMed website for Freeze8[17]. The resulting BCF files were split by study and consent group for distribution to approved dbGaP users. Quality control was performed centrally by the TOPMed IRC and the TOPMed Data Coordinating Center (DCC) as previously described[17]. Briefly, the two sequence quality criteria used in freeze 8 are: estimated DNA sample contamination below 10%, and 95% or more of the genome covered to 10× or greater. The variant filtering in TOPMed Freeze 8 is performed by (1) first calculating Mendelian consistency scores using known familial relatedness and duplicates, and (2) training a Support Vector Machine (SVM) classifier between known variant sites (positive labels) and Mendelian inconsistent variants. A small number of sex mismatches were detected as annotated females with low X and high Y chromosome depth or annotated males with high X and low Y chromosome depth. These samples were either excluded from the sample set to be released on dbGaP or their sample identities were resolved using information from prior array genotype comparisons and/or pedigree checks. Details regarding WGS data acquisition, processing and quality control vary among the TOPMed data freezes. Freeze-specific methods are described on the TOPMed website (https://www.nhlbiwgs.org/data-sets) and in documents included in each TOPMed accession released on dbGaP. The VCF/BCF files were converted to GDS (Genomic Data Structure) format by the DCC and were deposited into the dbGap TOPMed Exchange Area.

The genetic relationship matrix (GRM) is an N*N matrix of relatedness information of the samples included in the study and was computed centrally using 'PC-relate' R package (version: 1.24.0)[72]. Using the 'Genesis' R package (version:2.20.1)[73] we generated subsetted GRM for the samples with plasma lipid profiles. The GDS files with the variants were annotated internally by curating data from multiple database sources using Functional Annotation of Variant–Online Resource (FAVOR (http://favor.genohub.org)[13]. This study used the resultant aGDS (annotation GDS) files.

The MGB Biobank replication cohort was genotyped using three different arrays (Multiethnic Exome Global (Meg), Human multi-ethnic array (Mega), and Expanded multi-ethnic genotyping array (Megex)), and we separately imputed the data using TOPMed imputation server with default parameters[74,75]. This study applied the Version-r2 of the imputation panel, it includes 97,256 reference samples and ~300 M

genetic variants. The Illumina Global Screening array was used to genotype the Penn Medicine Biobank. Penn Medicine Biobank TOPMed imputation was performed using EAGLE[75] and Minimac[76] software. For this study, we downloaded variants that passed a min $R^2$ threshold of 0.3. The TOPMed imputation panel is robust, built from 97,256 deeply sequenced human genomes and contains 308,107,085 genetic variants from multi-ethnic samples. Imputation was performed in independent non-overlapping samples agnostic to phenotypes. The UKB imputed data was derived using merged UK10K[77], 1000 Genomes phase2 reference panels and was combined to the Haplotype reference Consortium[78] (HRC) using IMPUTE 4 program (https://jmarchini.org/software/). The UKB WGS data consist of whole genomes of 150,119 UKB participants with an average coverage of 32.5X. We used joint called VCFs from GraphTyper, which consist of 710,913,648 variants[79]. We used VCFs provided on the UK Biobank and conducted all the analysis in UKB Research Analysis Platform (UKB RAP).

## Single variant association

We performed genome-wide single variant association analyses for autosomal variants with minor allele frequency (MAF) >0.1% across the dataset with each of the four lipid phenotypes. We implemented the SAIGE-QT[80] method, which employs fast linear mixed models with kinship adjustment, in Encore (https://encore.sph.umich.edu/) for single variant association analyses. We additionally adjusted the model for covariates (PC1-PC11, age, sex, age², and study-groups [cohort-race subgrouping]).

We conducted single variant association replications for putative novel variants. After comparing the results with published lipid GWAS summary statistics, we filtered putative novel GWAS variants based on a stringent whole genome-wide significant threshold (alpha = $5 \times 10^{-9}$)[81]. Replication was performed in the MGB, Penn Medicine Biobanks and UK Biobank where linear regression models were fitted and adjusted for covariates as indicated above. In addition, we adjusted the MGB Biobank for study recruitment center and array and Penn Medicine Biobank for ancestry and BMI. In the MGB Biobank, we selected lipid concentrations closest to the sample acquisition time point and adjusted for statins if prescribed within one year prior to sample acquisition. In the Penn Biobank, we utilized each participant's median lipid concentration for replication; statins prescribed prior to lipid concentration used were adjusted in the models. In addition, we carried out meta-analysis using fixed effects model based on inverse-variance-weighted effect size for the two replication cohorts using METASOFT[82].

## Rare variant association test

We performed rare variant association (RVA) using the Variant-Set Test for Association using Annotation infoRmation (STAAR) pipeline[13,83]. STAARpipeline is a regression-based framework that permits adjustment of covariates, population structure, and relatedness by fitting linear and logistic mixed models for quantitative and dichotomous traits[83–85]. We chose STAAR to leverage the annotation information and associated scores that were available for TOPMed Freeze 8 data to incorporate the analysis of rare non-coding variants from whole genome sequencing. The method implements genome-wide scanning of rare variants (MAF <0.01) in gene-centric and region-based workflows. For each variant set, STAARpipeline calculates a set-based *p*-value using the STAAR method, which increases the analysis power by incorporating multiple in silico variant functional annotation scores capturing diverse genomic features and biochemical readouts[13]. We aggregated rare variants into multiple groups for coding and non-coding analyses. For the coding region, we defined five different aggregate masks of rare variants 1) plof (putative loss-of-function), plof-Ds (putative loss-of-function or disruptive missense), missense, disruptive-missense, and synonymous. For the non-coding regions, we used seven rare variant masks: (1) promoter-CAGE (promoter variants

within Cap Analysis of Gene Expression [CAGE] sites[86]), (2) promoter-DHS (promoter variants within DNase hypersensitivity [DHS] sites[87]), (3) enhancer-CAGE (enhancer within CAGE sites[88,89]), (4) enhancer-DHS (enhancer variants within DHS sites[87,89]), (5) UTR (rare variants in 3' untranslated region [UTR] and 5' UTR untranslated region), (6) upstream, and (7) downstream. Detailed explanations of the regions defined based on these masks is discussed within STAARpipeline[13,83].

In the gene-centric workflows, for both coding (within exonic boundaries) and non-coding (promoter: +/-3 kb window of transcription starting site (TSS), enhancer: GeneHancer predicted regions, UTR (both 5' and 3' UTR regions)/upstream/downstream: GENCODE Variant Effect Predictor (VEP) categories) regions, we considered only genes with at least two rare variants (i.e., 18,445 genes in all 22 autosomes). In the region-based workflows, we implemented two protocols: (1) a 'sliding window' approach, where we aggregated rare variants within 2-kb sliding windows and with 1-kb overlap length, and (2) a 'dynamic window' approach, where we executed SCANG[34] method and aggregated dynamically variant-sets between 40–300 variants per set, where the method systematically scans the whole genome with overlapping windows of varying sizes. The STAARpipeline R-package implements multiple rare-variants aggregate tests including SKAT[90], Burden[91] and ACAT[92] and integrates them as STAAR-O[13,83]. We performed gene-centric and region-based rare variant tests using annotated GDS files of TOPMed.

We completed aggregate tests as three-step process. In the first step, we fitted a null model using glmmkin() function. The null model was fitted for each of the four lipid phenotypes adjusted for all covariates and relatedness except the genotype of interest. In the second step, we ran genome-wide gene-centric and region-based rare-variant aggregate tests. The third step directed conditional analyses, where the results were adjusted for previously known significantly lipid-associated (i.e., $p < 5 \times 10^{-8}$ in external datasets) individual variants from GWAS Catalog[93] and Million Veterans Program (MVP)[15] GWAS summary statistics. To obtain effect estimates of significant aggregate sets, we associated the cumulative genotypes (binary scores) based on the variants forming the aggregates and used Glmm.Wald test from GMMAT R package[83](version 1.3.1). For significantly associated window-based rare variant aggregations, we trimmed the exonic variants and estimated the effects with only non-coding variants.

For the rare variant replication in UKB WGS data, we curated the rare variant aggregate sets in UKB RAP for the gene-centric coding/non-coding and region-based significant sets and applied STAAR workflow as demonstrated by the STAARpipeline (https://github.com/xihaoli/STAARpipeline) and describe above.

## Computational mining of single variant GWAS

**Gene-set enrichment using FUMA.** We performed enrichment analysis with single variant GWAS summary stats from the four lipids using FUMA[94] (version 1.3.7) with default parameters and significance at $5 \times 10^{-9}$. FUMA is an integrated platform which efficiently facilitates functional mapping and enrichment of GWAS-associated genes using multiple useful resources. The method uses 18 different biological data repositories and tools to process GWAS data. We additionally used MAGMA[95] (version 1.08) gene-based analysis enrichment workflow within FUMA with the complete GWAS summary data for eQTL based tissue enrichment. The functionally prioritized genes were visualized based on their protein-protein interaction networks using the STRING database[96].

***CETP* and *GAS6* gene expression and lipid trait colocalization.** We studied the correlation of LDL-C and HDL-C effects with eQTL effects at chromosome 16q13, which includes *CETP* and correlation of LDL-C and TC with eQTLs at rs7140110 of *GAS6*. We downloaded GTEx eQTL build 38 (version8) data for liver, adipose subcutaneous, and adipose visceral (omentum) tissues from GTEx on 16/APR/2020[97].

For the *CETP* variant analysis, we selected eQTLs with nominal significance (*p*-value < 0.05) and utilized the eQTL-gene pairs with the most significant *p*-values. Genes with at least 5 eQTLs were selected for the colocalization analysis. We selected variants with a suggestive significance (*p*-value $<5 \times 10^{-7}$) for LDL-C or HDL-C effects within 500 kb of the lead locus variant. For the *GAS6* variant analysis, we curated all the GWAS variants within 500 kb of the lead variant with nominal significance (*p*-value < 0.05) and matched them to eQTL data where the transcription starting site of the corresponding gene is within +/−500 kb. We conducted colocalization analysis using the coloc.abf() function[98] and identified nominally significant (PP.H4 > $1 \times 10^{-03}$) genes-eQTL pairs. The coloc methodology implements an efficient statistical framework to identify shared variants from two association signals through posteriors probabilities. Finally, we used the colocalized signals and compared the significant genes using STRING[96], a protein-protein interaction database. All the correlation tests were conducted in R, where we calculated Pearson correlations between the lipid effect estimates and gene expression effects (slope) from GTEx.

**Phenome wide association analysis.** The complex trait information was curated from UK Biobank resource, where we curated multiple disease phenotypes for UKB samples into International Classification of Diseases (ICD)-based phecodes based on phecode map (https://phewascatalog.org) using the PheWAS R package (version PheWAS_0.99.5-4). We conducted a phenome-wide association analysis (PheWAS) using a logistic regression model glm() in R. We adjusted the models for PC1–10, age, age[2], sex, and race.

## Calculation of heritability estimates from TOPMed WGS data

We calculated heritabilities estimated for the four lipids using TOPMed WGS data using Greml-LDMS approach[39], where we binned the variants into four MAF bins based on minor allele frequency and grouped the variants to four LD quartiles based on LD score calculated by GCTA method[99]. The four MAF bins used in this study includes >=0.05, >=0.01 to <0.05, >=0.001 to <0.01 and >=0.0001 to <0.001. We excluded any variant with MAF < 0.0001 from this analysis. The hereditary estimation was calculated for three ancestral groups (African, European, Hispanic) where only unrelated samples (kinship score < 0.025) were included in the analysis. We excluded the other two ancestral groups (i.e., Asian and Samoan) from this analysis due to insufficient sample sizes. In total we included 9640, 21568 and 10631 in African, European and Hispanic ancestries respectively. For each MAF bin, we implemented certain quality control (QC) measures using PLINK software[20], which includes; genotype missingness (--geno 0.05), sample missingness (--mind 0.05), Hardy-Weinberg equilibrium (--hwe $10^{-6}$) and LD pruned variants (--indep-pairwise 50 5 0.1) as implemented by Wainschtein et al.[39]. Next, we implemented Greml-LDMS with LD score region as 200 and GRM cut-off as 0.05 for the four lipid phenotypes. We calculated 20 principal components from the QC passed variants in each MAF bin and implemented GCTA workflow with --reml-no-constrain, --reml-no-lrt and --reml-maxit 10,000 parameters to avoid the no-convergence issues and negative $h^2$ estimates. For comparing the $h^2$ estimates between variants from WGS data and array-genotypes, first, we used QC passed WGS variants as mentioned above, second, we curated the variants from MGB Biobank array data and intersected them with WGS variants from TOPMed. Next, we calculated heritability estimates for array-genotype variants and compared with $h^2$ estimates from WGS variants for the three ancestral groups.

## Reporting summary

Further information on research design is available in the Nature Research Reporting Summary linked to this article.

## Data availability

Individual whole-genome sequence data for TOPMed and harmonized lipids at individual sample level are available through restricted access via the TOPMed dbGaP Exchange area. Summary level genotype data from TOPMed are available through the BRAVO browser (https://bravo.sph.umich.edu/). The UK Biobank (UKB) whole-genome sequence data can be accessed through UKB Research Analysis Platform (RAP), through the UKB approval system (https://www.ukbiobank.ac.uk). The Mass General Brigham Biobank (MGBB) individual-level data are available from https://personalizedmedicine.partners.org/Biobank/Default.aspx, where the data is available through institutional review board (IRB) approval, therefore not publicly available. Individual-level data from Penn Medicine BioBank is not publicly available due to research participants privacy concerns. The summary data captured using whole exome sequencing can be accessed through PMBB Genome Browser (https://pmbb.med.upenn.edu/allele-frequency/). The dbGaP accessions for TOPMed cohorts are as follows: Old Order Amish (Amish) *phs000956 and phs00039;* Atherosclerosis Risk in Communities study (ARIC) *phs001211 and phs000280;* Mt Sinai BioMe Biobank (BioMe) *phs001644 and phs000925;* Coronary Artery Risk Development in Young Adults (CARDIA) phs001612 and phs000285; Cleveland Family Study (CFS) *phs000954 and phs000284;* Cardiovascular Health Study (CHS) *phs001368 and phs000287;* Diabetes Heart Study (DHS) *phs001412 and phs001012;* Framingham Heart Study (FHS) *phs000974 and phs000007;* Genetic Studies of Atherosclerosis Risk (GeneSTAR) *phs001218 and phs000375;* Genetic Epidemiology Network of Arteriopathy (GENOA) *phs001345 and phs001238;* Genetic Epidemiology Network of Salt Sensitivity (GenSalt) *phs001217 and phs000784;* Genetics of Lipid-Lowering Drugs and Diet Network (GOLDN) *phs001359 and phs000741;* Hispanic Community Health Study - Study of Latinos (HCHS_SOL) *phs001395 and phs000810;* Hypertension Genetic Epidemiology Network and Genetic Epidemiology Network of Arteriopathy (HyperGEN) *phs001293 and phs001293;* Jackson Heart Study (JHS) *phs000964 and phs000286;* Multi-Ethnic Study of Atherosclerosis (MESA) *phs001416 and phs000209;* Massachusetts General Hospital Atrial Fibrillation Study (MGH_AF) *phs001062 and phs001001;* San Antonio Family Study (SAFS) *phs001215 and phs000462;* Samoan Adiposity Study (SAS) *phs000972 and phs000914;* Taiwan Study of Hypertension using Rare Variants (THRV) *phs001387 and phs001387;* Women's Health Initiative (WHI) *phs001237 and phs000200.*

## Code availability

Codes used to implement STAAR workflows are available at https://github.com/xihaoli/STAAR and https://github.com/xihaoli/STAARpipeline. Workflow implemented for whole genome heritability calculations are available at https://github.com/CNSGenomics/Heritability_WGS.

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

## Acknowledgements

Whole genome sequencing (WGS) for the Trans-Omics in Precision Medicine (TOPMed) program was supported by the National Heart, Lung and Blood Institute (NHLBI). P.N. is supported by grants from the National Heart, Lung, and Blood Institute (R01HL142711, R01HL148050, R01HL151283, R01HL148565, R01HL135242, R01HL151152), Fondation Leducq (TNE-18CVD04), and Massachusetts General Hospital (Paul and Phyllis Fireman Endowed Chair in Vascular Medicine). G.M.P. is supported by NIH grants R01HL142711 and R01HL127564. X.Lin is supported by grants R35-CA197449, U19-CA203654, R01-HL113338, and U01-HG009088. Prior to his employment at Novartis and during this work S.A.L. was supported by NIH grants R01HL139731, R01HL157635, and American Heart Association 18SFRN34250007. We like to acknowledge all the grants that supported this study, R01 HL121007, U01 HL072515, R01 AG18728, X01HL134588, HL 046389, HL113338, and 1R35HL135818, K01 HL135405, R03 HL154284, U01HL072507, R01HL087263, R01HL090682, P01HL045522, R01MH078143, R01MH078111, R01MH083824, U01DK085524, R01HL113323, R01HL093093, R01HL140570, R01HL142711, R01HL127564, R01HL148050, R01HL148565, HL105756, and Leducq TNE-18CVD04. The views expressed in this manuscript are those of the authors and do not necessarily represent the views of the National Heart, Lung, and Blood Institute; the National Institutes of Health; or the U.S.Department of Health and Human Services. Detailed acknowledgements provided in Supplementary Note 2.

## Author contributions

M.S.S., G.M.P., and P.N. designed the study. M.S.S. carried out all the primary analysis with critical inputs from G.M.P. and P.N. M.S.S., Xih.L, Z.L., A.P., D.Y.Z., J.P., S.A., J.C.B., J.A.B., B.E.C., L.M.C., R.H.C., J.E.C., L.F., P.S.V., R.D., B.I.F., M.G., X.G., N.H.C., B.H., C.M.H., M.R.I., T.N.K., B.G.K., L.L., Xia.L, M.L., S.A.L., A.W.M., P.M., M.E.M., A.C.M., T.N., J.R.O.C., N.D.P., P.A.P., M.S.R., J.A.S., X.S., K.D.T., R.P.T., M.Y.T., Z.W., Y.W., B.W., J.T.W., L.R.Y., W.Z., D.K.A., J. Blangero, E.B., D.W.B., Y.I.C., A.C., L.A.C., S.K.D., P.T.E., M.F., S. Gabriel, S. Germer, R.G., J.H., R.C.K., S.L.R.K., R. Kim, C.K., R.J.F.L., K.M., R.A.M., S.T.M., B.D.M., D.N., K.E.N., B.M.P., S. Redline, A.P.R., R.S.V., S.S.R., C.W., J.I.R., D.J.R., X.Lin., G.M.P., and P.N. acquired, analyzed or interpreted data. M.S.S., G.M.P. and P.N. wrote the first draft of the manuscript and all others provided intellectual revisions. G.M.P. and P.N. and NHLBI TOPMed Lipids Working Group provided administrative, technical, or material support.

## Competing interests

P.N. reports investigator-initiated grant support from Amgen, Apple, AstraZeneca, and Boston Scientific, personal fees from Apple, AstraZeneca, Blackstone Life Sciences, Foresite Labs, Genentech, TenSixteen Bio, and Novartis, scientific advisory board membership of geneXwell and TenSixteen Bio, and spousal employment at Vertex, all unrelated to the present work. B.P. serves on the Steering Committee of the Yale Open Data Access Project funded by Johnson & Johnson.

M.E.M. receives funding from Regeneron Pharmaceutical Inc. unrelated to this work. S.A. has employment and equity in 23andMe, Inc. The spouse of C.J.W. works at Regeneron. S.A.L. is a full-time employee of Novartis as of July 18, 2022. S.A.L. has received sponsored research support from Bristol Myers Squibb, Pfizer, Boehringer Ingelheim, Fitbit, Medtronic, Premier, and IBM, and has consulted for Bristol Myers Squibb, Pfizer, Blackstone Life Sciences, and Invitae. X. Lin is a consultant of AbbVie Pharmaceuticals and Verily Life Sciences. The remaining authors declare no competing interests.

## Additional information

**Margaret Sunitha Selvaraj** [1,2,3], **Xihao Li** [4], **Zilin Li** [4], **Akhil Pampana** [2], **David Y. Zhang** [5,6], **Joseph Park** [5,6], **Stella Aslibekyan** [7], **Joshua C. Bis** [8], **Jennifer A. Brody** [8], **Brian E. Cade** [9], **Lee-Ming Chuang** [10], **Ren-Hua Chung** [11], **Joanne E. Curran** [12], **Lisa de las Fuentes** [13,14], **Paul S. de Vries** [15], **Ravindranath Duggirala** [12], **Barry I. Freedman** [16], **Mariaelisa Graff** [17], **Xiuqing Guo** [18], **Nancy Heard-Costa** [19], **Bertha Hidalgo** [7], **Chii-Min Hwu** [20], **Marguerite R. Irvin** [7], **Tanika N. Kelly** [21,22], **Brian G. Kral** [23], **Leslie Lange** [24], **Xiaohui Li** [18], **Martin Lisa** [25], **Steven A. Lubitz** [1,26], **Ani W. Manichaikul** [27], **Preuss Michael** [28], **May E. Montasser** [29], **Alanna C. Morrison** [15], **Take Naseri** [30], **Jeffrey R. O'Connell** [29], **Nicholette D. Palmer** [31], **Patricia A. Peyser** [32], **Muagututia S. Reupena** [33], **Jennifer A. Smith** [32], **Xiao Sun** [21], **Kent D. Taylor** [18], **Russell P. Tracy** [34], **Michael Y. Tsai** [35], **Zhe Wang** [28], **Yuxuan Wang** [36], **Wei Bao** [37], **John T. Wilkins** [38], **Lisa R. Yanek** [23], **Wei Zhao** [32], **Donna K. Arnett** [39], **John Blangero** [12], **Eric Boerwinkle** [15], **Donald W. Bowden** [31], **Yii-Der Ida Chen** [40], **Adolfo Correa** [41], **L. Adrienne Cupples** [36], **Susan K. Dutcher** [42], **Patrick T. Ellinor** [1,26], **Myriam Fornage** [43], **Stacey Gabriel** [44], **Soren Germer** [45], **Richard Gibbs** [46], **Jiang He** [21,22], **Robert C. Kaplan** [47,48], **Sharon L. R. Kardia** [32], **Ryan Kim** [49], **Charles Kooperberg** [48], **Ruth J. F. Loos** [28,50], **Karine A Viaud-Martinez** [51], **Rasika A. Mathias** [23], **Stephen T. McGarvey** [52], **Braxton D. Mitchell** [29,53], **Deborah Nickerson** [54], **Kari E. North** [17], **Bruce M. Psaty** [8,55,56], **Susan Redline** [9], **Alexander P. Reiner** [55,48], **Ramachandran S. Vasan** [57,58,59], **Stephen S. Rich** [27], **Cristen Willer** [60], **Jerome I. Rotter** [18], **Daniel J. Rader** [5,6,61], **Xihong Lin** [2,4,62], **NHLBI Trans-Omics for Precision Medicine (TOPMed) Consortium**\*, **Gina M. Peloso** [36,220] ✉ & **Pradeep Natarajan** [1,2,3,220] ✉

[1]Cardiovascular Research Center, Massachusetts General Hospital, Boston, MA 02114, USA. [2]Program in Medical and Population Genetics, Broad Institute of Harvard and MIT, Cambridge, MA 02142, USA. [3]Department of Medicine, Harvard Medical School, Boston, MA 02115, USA. [4]Department of Biostatistics, Harvard T.H. Chan School of Public Health, Boston, MA 02115, USA. [5]Department of Genetics, Perelman School of Medicine, University of Pennsylvania, Philadelphia, PA 19104, USA. [6]Department of Medicine, Perelman School of Medicine, University of Pennsylvania, Philadelphia, PA 19104, USA. [7]Department of Epidemiology, University of Alabama at Birmingham School of Public Health, Birmingham, AL, USA. [8]Cardiovascular Health Research Unit, Department of Medicine, University of Washington, Seattle, WA, USA. [9]Department of Medicine, Brigham and Women's Hospital, Harvard Medical School, Boston, MA, USA. [10]Department of Internal Medicine, National Taiwan University Hospital, Taipei, Taiwan. [11]Institute of Population Health Sciences, National Health Research Institutes, Zhunan 350, Taiwan. [12]Department of Human Genetics and South Texas Diabetes and Obesity Institute, University of Texas Rio Grande Valley School of Medicine, Brownsville, TX 78520, USA. [13]Department of Medicine, Cardiovascular Division, Washington University School of Medicine, St. Louis, MO, USA. [14]Division of Biostatistics, Washington University School of Medicine, St. Louis, MO, USA. [15]Human Genetics Center, Department of Epidemiology, Human Genetics, and Environmental Sciences, School of Public Health, The University of Texas Health Science Center at Houston, Houston, TX, USA. [16]Department of Internal Medicine, Section on Nephrology, Wake Forest School of Medicine, Winston-Salem, NC 27157, USA. [17]Department of Epidemiology, UNC Chapel Hill, Chapel Hill, NC, USA. [18]The Institute for Translational Genomics and Population Sciences, Department of Pediatrics, The Lundquist Institute for Biomedical Innovation at Harbor-UCLA Medical Center, Torrance, CA, USA. [19]Department of Neurology, Boston university School of Medicine, Boston, MA, USA. [20]Section of Endocrinology and Metabolism, Department of Medicine, Taipei Veterans General Hospital, Taipei, Taiwan. [21]Department of Epidemiology, Tulane University School of Public Health and Tropical Medicine, New Orleans, LA 70112, USA. [22]Tulane University Translational Science Institute, New Orleans, LA 70112, USA. [23]Department of Medicine, Johns Hopkins University School of Medicine, Baltimore, MD 21205, USA. [24]Division of Biomedical

Informatics and Personalized Medicine, Department of Medicine, University of Colorado Anschutz Medical Campus, Aurora, CO, USA. [25]Department of Medicine, George Washington University, Washingron, DC, USA. [26]Cardiovascular Disease Initiative, The Broad Institute of MIT and Harvard, Cambridge, MA 02124, USA. [27]Department of Public Health Sciences, Center for Public Health Genomics, University of Virginia, Charlottesville, VA, USA. [28]The Charles Bronfman Institute for Personalized Medicine, Icahn School of Medicine at Mount Sinai, New York, NY, USA. [29]Department of Medicine, University of Maryland School of Medicine, Baltimore, MD, USA. [30]Ministry of Health, Government of Samoa, Samoa, USA. [31]Department of Biochemistry, Wake Forest School of Medicine, Winston-Salem, NC 27157, USA. [32]Department of Epidemiology, University of Michigan, Ann Arbor, MI 48109, USA. [33]Lutia i Puava ae Mapu i Fagalele, Apia, Samoa. [34]Departments of Pathology & Laboratory Medicine and Biochemistry, Larner College of Medicine at the University of Vermont, Colchester, VT, USA. [35]Department of Laboratory Medicine and Pathology, University of Minneosta, Minneapolis, MN, USA. [36]Department of Biostatistics, Boston University School of Public Health, Boston, MA 02118, USA. [37]Institute of Public Health, Division of Life Sciences and Medicine, University of Science and Technology of China, Hefei, Anhui 230026, China. [38]Department of Medicine (Cardiology) and Department of Preventive Medicine, Northwestern University Feinberg School of Medicine, Chicago, IL, USA. [39]Dean's Office, University of Kentucky College of Public Health, Lexington, KY, USA. [40]Lundquist Institute for Biomedical Innovation at Harbor-UCLA Medical Center, Torrance, CA, USA. [41]Department of Population Health Science, University of Mississippi Medical Center, Jackson, MS, USA. [42]The McDonnell Genome Institute, Washington University School of Medicine, St. Louis, MO 63108, USA. [43]Brown Foundation Institute of Molecular Medicine, McGovern Medical School, The University of Texas Health Science Center at Houston, Houston, TX 7722, USA. [44]Broad Institute, Cambridge, MA 02142, USA. [45]New York Genome Center, New York, NY 10013, USA. [46]Baylor College of Medicine Human Genome Sequencing Center, Houston, TX 77030, USA. [47]Department of Epidemiology and Population Health, Albert Einstein College of Medicine, Bronx, NY 10461, USA. [48]Division of Public Health Sciences, Fred Hutchinson Cancer Research Center, Seattle, WA 98109, USA. [49]Psomagen, Inc. (formerly Macrogen USA), Rockville, MD, USA. [50]NNF Center for Basic Metabolic Research, University of Copenhagen, Cophenhagen, Denmark. [51]Illumina Laboratory Services, Illumina. Inc, San Diego 92122, USA. [52]Department of Epidemiology, International Health Institute, Brown University, Providence, RI, USA. [53]Geriatrics Research and Education Clinical Center, Baltimore Veterans Administration Medical Center, Baltimore, MD, USA. [54]University of Washington, Department of Genome Sciences, Seattle, WA 98195, USA. [55]Department of Epidemiology, University of Washington, Seattle, WA, USA. [56]Department of Health Systems and Population Health, University of Washington, Seattle, WA, USA. [57]Sections of Preventive medicine and Epidemiology, Cardiovascular medicine, Department of Medicine, Boston University School of Medicine, Boston, MA, USA. [58]Department of Epidemiology, Boston University School of Public Health, Boston, MA, USA. [59]Framingham Heart Study, Framingham, MA, USA. [60]University of Michigan, Internal Medicine, Ann Arbor, MI 48109, USA. [61]Institute for Translational Medicine and Therapeutics, Perelman School of Medicine, University of Pennsylvania, Philadelphia, PA 19104, USA. [62]Department of Statistics, Harvard University, Cambridge, MA 02138, USA. [220]These authors jointly supervised this work: Gina M. Peloso, Pradeep Natarajan. *A list of authors and their affiliations appears at the end of the paper. ✉e-mail: gpeloso@bu.edu; pnatarajan@mgh.harvard.edu

## NHLBI Trans-Omics for Precision Medicine (TOPMed) Consortium

Namiko Abe[63], Gonçalo Abecasis[64], Francois Aguet[65], Christine Albert[66], Laura Almasy[67], Alvaro Alonso[68], Seth Ament[69], Peter Anderson[70], Pramod Anugu[71], Deborah Applebaum-Bowden[72], Kristin Ardlie[65], Dan Arking[73], Allison Ashley-Koch[74], Tim Assimes[75], Paul Auer[76], Dimitrios Avramopoulos[73], Najib Ayas[77], Adithya Balasubramanian[78], John Barnard[79], Kathleen Barnes[80], R. Graham Barr[81], Emily Barron-Casella[73], Lucas Barwick[82], Terri Beaty[73], Gerald Beck[83], Diane Becker[84], Lewis Becker[73], Rebecca Beer[85], Amber Beitelshees[69], Emelia Benjamin[86], Takis Benos[87], Marcos Bezerra[88], Larry Bielak[64], Thomas Blackwell[64], Russell Bowler[89], Ulrich Broeckel[90], Jai Broome[70], Deborah Brown[91], Karen Bunting[63], Esteban Burchard[92], Carlos Bustamante[93], Erin Buth[94], Jonathan Cardwell[95], Vincent Carey[96], Julie Carrier[97], Cara Carty[98], Richard Casaburi[99], Juan P. Casas Romero[100], James Casella[73], Peter Castaldi[101], Mark Chaffin[65], Christy Chang[69], Yi-Cheng Chang[102], Daniel Chasman[103], Sameer Chavan[95], Bo-Juen Chen[63], Wei-Min Chen[104], Yii-Der Ida Chen[105], Michael Cho[96], Seung Hoan Choi[65], Mina Chung[106], Clary Clish[107], Suzy Comhair[108], Matthew Conomos[94], Elaine Cornell[109], Carolyn Crandall[99], James Crapo[110], L. Adrienne Cupples[111], Jeffrey Curtis[64], Brian Custer[112], Coleen Damcott[69], Dawood Darbar[113], Sean David[114], Colleen Davis[70], Michelle Daya[95], Mariza de Andrade[115], Michael DeBaun[116], Ranjan Deka[117], Dawn DeMeo[96], Scott Devine[69], Huyen Dinh[78], Harsha Doddapaneni[78], Qing Duan[118], Shannon Dugan-Perez[78], Ravi Duggirala[119], Jon Peter Durda[109], Charles Eaton[120], Lynette Ekunwe[71], Adel El Boueiz[121], Leslie Emery[70], Serpil Erzurum[79], Charles Farber[104], Jesse Farek[78], Tasha Fingerlin[122], Matthew Flickinger[64], Nora Franceschini[123], Chris Frazar[70], Mao Fu[69], Stephanie M. Fullerton[70], Lucinda Fulton[124], Weiniu Gan[85], Shanshan Gao[95], Yan Gao[71], Margery Gass[125], Heather Geiger[126], Bruce Gelb[127], Mark Geraci[128], Robert Gerszten[129], Auyon Ghosh[96], Chris Gignoux[75], Mark Gladwin[87], David Glahn[130], Stephanie Gogarten[70], Da-Wei Gong[69], Harald Goring[131], Sharon Graw[80], Kathryn J. Gray[132], Daniel Grine[95], Colin Gross[64], C. Charles Gu[124], Yue Guan[69], Namrata Gupta[65], David M. Haas[133], Jeff Haessler[125], Michael Hall[134], Yi Han[78], Patrick Hanly[135], Daniel Harris[136], Nicola L. Hawley[137], Ben Heavner[94], Susan Heckbert[138], Ryan Hernandez[92], David Herrington[139], Craig Hersh[140], Bertha Hidalgo[141], James Hixson[142], Brian Hobbs[96], John Hokanson[95], Elliott Hong[69], Karin Hoth[143], Chao Agnes Hsiung[144], Jianhong Hu[78], Yi-Jen Hung[145], Haley Huston[146], Chii Min Hwu[147], Rebecca Jackson[148], Deepti Jain[70], Cashell Jaquish[85], Jill Johnsen[149], Andrew Johnson[85], Craig Johnson[70], Rich Johnston[68], Kimberly Jones[73], Hyun Min Kang[150], Shannon Kelly[151], Eimear Kenny[127], Michael Kessler[69], Alyna Khan[70], Ziad Khan[78], Wonji Kim[152], John Kimoff[153], Greg Kinney[154], Barbara Konkle[146], Holly Kramer[155], Christoph Lange[156], Ethan Lange[95], Cathy Laurie[70], Cecelia Laurie[70], Meryl LeBoff[96], Jiwon Lee[96], Sandra Lee[78], Wen-Jane Lee[147], Jonathon LeFaive[64], David Levine[70], Dan Levy[85], Joshua Lewis[69], Yun Li[118], Henry Lin[105], Honghuang Lin[157], Simin Liu[158], Yongmei Liu[159], Yu Liu[160],

Kathryn Lunetta[157], James Luo[85], Ulysses Magalang[161], Michael Mahaney[162], Barry Make[73], Alisa Manning[163], JoAnn Manson[96], Lisa Martin[164], Melissa Marton[126], Susan Mathai[95], Susanne May[94], Patrick McArdle[69], Merry-Lynn McDonald[141], Sean McFarland[152], Daniel McGoldrick[165], Caitlin McHugh[94], Becky McNeil[166], Hao Mei[71], James Meigs[167], Vipin Menon[78], Luisa Mestroni[80], Ginger Metcalf[78], Deborah A. Meyers[168], Emmanuel Mignot[169], Julie Mikulla[85], Nancy Min[71], Mollie Minear[170], Ryan L. Minster[87], Matt Moll[101], Zeineen Momin[78], Courtney Montgomery[171], Donna Muzny[78], Josyf C. Mychaleckyj[104], Girish Nadkarni[127], Rakhi Naik[73], Sergei Nekhai[172], Sarah C. Nelson[94], Bonnie Neltner[95], Caitlin Nessner[78], Osuji Nkechinyere[78], Jeff O'Connell[173], Tim O'Connor[69], Heather Ochs-Balcom[174], Geoffrey Okwuonu[78], Allan Pack[175], David T. Paik[176], James Pankow[177], George Papanicolaou[85], Cora Parker[178], Juan Manuel Peralta[119], Marco Perez[75], James Perry[69], Ulrike Peters[179], Lawrence S. Phillips[68], Jacob Pleiness[64], Toni Pollin[69], Wendy Post[180], Julia Powers Becker[181], Meher Preethi Boorgula[95], Michael Preuss[127], Pankaj Qasba[85], Dandi Qiao[96], Zhaohui Qin[68], Nicholas Rafaels[182], Laura Raffield[183], Mahitha Rajendran[78], Ramachandran S. Vasan[157], D. C. Rao[124], Laura Rasmussen-Torvik[184], Aakrosh Ratan[104], Robert Reed[69], Catherine Reeves[185], Elizabeth Regan[110], Alex Reiner[186], Muagututia S. Reupena[33], Ken Rice[70], Rebecca Robillard[187], Nicolas Robine[126], Dan Roden[188], Carolina Roselli[65], Ingo Ruczinski[73], Alexi Runnels[126], Pamela Russell[95], Sarah Ruuska[146], Kathleen Ryan[69], Ester Cerdeira Sabino[189], Danish Saleheen[190], Shabnam Salimi[69], Sejal Salvi[78], Steven Salzberg[73], Kevin Sandow[191], Vijay G. Sankaran[192], Jireh Santibanez[78], Karen Schwander[124], David Schwartz[95], Frank Sciurba[87], Christine Seidman[193], Jonathan Seidman[194], Frédéric Sériès[195], Vivien Sheehan[196], Stephanie L. Sherman[197], Amol Shetty[69], Aniket Shetty[95], Wayne Hui-Heng Sheu[147], M. Benjamin Shoemaker[198], Brian Silver[199], Edwin Silverman[96], Robert Skomro[200], Albert Vernon Smith[201], Josh Smith[70], Nicholas Smith[138], Tanja Smith[63], Sylvia Smoller[202], Beverly Snively[203], Michael Snyder[75], Tamar Sofer[96], Nona Sotoodehnia[70], Adrienne M. Stilp[70], Garrett Storm[204], Elizabeth Streeten[69], Jessica Lasky Su[96], Yun Ju Sung[124], Jody Sylvia[96], Adam Szpiro[70], Daniel Taliun[64], Hua Tang[205], Margaret Taub[73], Matthew Taylor[80], Simeon Taylor[69], Marilyn Telen[74], Timothy A. Thornton[70], Machiko Threlkeld[206], Lesley Tinker[125], David Tirschwell[70], Sarah Tishkoff[207], Hemant Tiwari[208], Catherine Tong[209], Dhananjay Vaidya[73], David Van Den Berg[210], Peter VandeHaar[64], Scott Vrieze[177], Tarik Walker[95], Robert Wallace[143], Avram Walts[95], Fei Fei Wang[70], Heming Wang[211], Jiongming Wang[201], Karol Watson[99], Jennifer Watt[78], Daniel E. Weeks[87], Joshua Weinstock[150], Bruce Weir[70], Scott T. Weiss[212], Lu-Chen Weng[213], Jennifer Wessel[214], Kayleen Williams[94], L. Keoki Williams[215], Carla Wilson[96], James Wilson[216], Lara Winterkorn[126], Quenna Wong[70], Joseph Wu[176], Huichun Xu[69], Ivana Yang[95], Ketian Yu[64], Seyedeh Maryam Zekavat[65], Yingze Zhang[217], Snow Xueyan Zhao[110], Wei Zhao[218], Xiaofeng Zhu[219], Michael Zody[63] & Sebastian Zoellner[64]

[63]New York Genome Center, New York, NY 10013, USA. [64]University of Michigan, Ann Arbor, MI 48109, USA. [65]Broad Institute, Cambridge, MA 02142, USA. [66]Cedars Sinai, Boston, MA 02114, USA. [67]Children's Hospital of Philadelphia, University of Pennsylvania, Philadelphia, PA 19104, USA. [68]Emory University, Atlanta, GA 30322, USA. [69]University of Maryland, Baltimore, MD 21201, USA. [70]University of Washington, Seattle, WA 98195, USA. [71]University of Mississippi, Jackson, MS 38677, USA. [72]National Institutes of Health, Bethesda, MD 20892, USA. [73]Johns Hopkins University, Baltimore, MD 21218, USA. [74]Duke University, Durham, NC 27708, USA. [75]Stanford University, Stanford, CA 94305, USA. [76]University of Wisconsin Milwaukee, Milwaukee, WI 53211, USA. [77]Providence Health Care, Medicine, Vancouver, USA. [78]Baylor College of Medicine Human Genome Sequencing Center, Houston, TX 77030, USA. [79]Cleveland Clinic, Cleveland, OH 44195, USA. [80]University of Colorado Anschutz Medical Campus, Aurora, CO 80045, USA. [81]Columbia University, New York, NY 10032, USA. [82]The Emmes Corporation, LTRC, Rockville, MD 20850, USA. [83]Cleveland Clinic, Quantitative Health Sciences, Cleveland, OH 44195, USA. [84]Johns Hopkins University, Medicine, Baltimore, MD 21218, USA. [85]National Heart, Lung, and Blood Institute, National Institutes of Health, Bethesda, MD 20892, USA. [86]Boston University, Massachusetts General Hospital, Boston University School of Medicine, Boston, MA 02114, USA. [87]University of Pittsburgh, Pittsburgh, PA 15260, USA. [88]Fundação de Hematologia e Hemoterapia de Pernambuco - Hemope, Recife 52011-000, Brazil. [89]National Jewish Health, National Jewish Health, Denver, CO 80206, USA. [90]Medical College of Wisconsin, Milwaukee, WI 53226, USA. [91]University of Texas Health at Houston, Pediatrics, Houston, TX 77030, USA. [92]University of California, San Francisco, San Francisco, CA 94143, USA. [93]Stanford University, Biomedical Data Science, Stanford, CA 94305, USA. [94]University of Washington, Biostatistics, Seattle, WA 98195, USA. [95]University of Colorado at Denver, Denver, CO 80204, USA. [96]Brigham & Women's Hospital, Boston, MA 02115, USA. [97]University of Montreal, Quebec, Canada. [98]Washington State University, Pullman, WA 99164, USA. [99]University of California, Los Angeles, Los Angeles, CA 90095, USA. [100]Brigham & Women's Hospital, Boston, USA. [101]Brigham & Women's Hospital, Medicine, Boston, MA 02115, USA. [102]National Taiwan University, Taipei 10617, Taiwan. [103]Brigham & Women's Hospital, Division of Preventive Medicine, Boston, MA 02215, USA. [104]University of Virginia, Charlottesville, VA 22903, USA. [105]Lundquist Institute, Torrance, CA 90502, USA. [106]Cleveland Clinic, Cleveland Clinic, Cleveland, OH 44195, USA. [107]Broad Institute, Metabolomics Platform, Cambridge, MA 02142, USA. [108]Cleveland Clinic, Immunity and Immunology, Cleveland, OH 44195, USA. [109]University of Vermont, Burlington, VT 05405, USA. [110]National Jewish Health, Denver, CO 80206, USA. [111]Boston University, Biostatistics, Boston, MA 02115, USA. [112]Vitalant Research Institute, San Francisco, CA 94118, USA. [113]University of Illinois at Chicago, Chicago, IL 60607, USA. [114]University of Chicago, Chicago, IL 60637, USA. [115]Mayo Clinic, Health Quantitative Sciences Research, Rochester, MN 55905, USA. [116]Vanderbilt University, Nashville, TN 37235, USA. [117]University of Cincinnati, Cincinnati, Ohio 45220, USA. [118]University of North Carolina, Chapel Hill, NC 27599, USA. [119]University of Texas Rio Grande Valley School of Medicine, Edinburg, TX 78539, USA. [120]Brown University, Providence, RI 02912, USA. [121]Harvard University, Channing Division of Network Medicine, Cambridge, MA 02138, USA. [122]National Jewish Health, Center for Genes, Environment and Health, Denver, CO 80206, USA. [123]University of North Carolina, Epidemiology, Chapel Hill, NC 27599, USA. [124]Washington University in St Louis, St Louis, MO 63130, USA. [125]Fred Hutchinson Cancer Research Center, Seattle, WA 98109, USA. [126]New York Genome Center, New York City, NY 10013, USA. [127]Icahn School of Medicine at Mount Sinai, New York, NY 10029, USA. [128]University of Pittsburgh, Pittsburgh, PA, USA. [129]Beth Israel Deaconess Medical Center, Boston, MA 02215, USA. [130]Boston Children's Hospital, Harvard Medical School, Department of Psychiatry, Boston, MA 02115, USA. [131]University of Texas Rio Grande Valley School of Medicine, San Antonio,

TX 78229, USA. [132]Mass General Brigham, Obstetrics and Gynecology, Boston, MA 02115, USA. [133]Indiana University, OB/GYN, Indianapolis, Indiana 46202, USA. [134]University of Mississippi, Cardiology, Jackson, MS 39216, USA. [135]University of Calgary, Medicine, Calgary, Canada. [136]University of Maryland, Genetics, Philadelphia, PA 19104, USA. [137]Yale University, Department of Chronic Disease Epidemiology, Connecticut 06520, USA. [138]University of Washington, Epidemiology, Seattle, WA 98195, USA. [139]Wake Forest Baptist Health, Winston-Salem, NC 27157, USA. [140]Brigham & Women's Hospital, Channing Division of Network Medicine, Boston, MA 02115, USA. [141]University of Alabama, Birmingham, AL 35487, USA. [142]University of Texas Health at Houston, Houston, TX 77225, USA. [143]University of Iowa, Iowa City, IA 52242, USA. [144]National Health Research Institute Taiwan, Institute of Population Health Sciences, NHRI, Miaoli County 350, Taiwan. [145]Tri-Service General Hospital National Defense Medical Center, Taipei, Taiwan. [146]Blood Works Northwest, Seattle, WA 98104, USA. [147]Taichung Veterans General Hospital Taiwan, Taichung City 407, Taiwan. [148]Oklahoma State University Medical Center, Internal Medicine, DIvision of Endocrinology, Diabetes and Metabolism, Columbus, OH 43210, USA. [149]Blood Works Northwest, Research Institute, Seattle, WA 98104, USA. [150]University of Michigan, Biostatistics, Ann Arbor, MI 48109, USA. [151]University of California, San Francisco, San Francisco, CA 94118, USA. [152]Harvard University, Cambridge, MA 02138, USA. [153]McGill University, Montréal, QC H3A 0G4, Canada. [154]University of Colorado at Denver, Epidemiology, Aurora, CO 80045, USA. [155]Loyola University, Public Health Sciences, Maywood, IL 60153, USA. [156]Harvard School of Public Health, Biostats, Boston, MA 02115, USA. [157]Boston University, Boston, MA 02215, USA. [158]Brown University, Epidemiology and Medicine, Providence, RI 02912, USA. [159]Duke University, Cardiology, Durham, NC 27708, USA. [160]Stanford University, Cardiovascular Institute, Stanford, CA 94305, USA. [161]Ohio State University, Division of Pulmonary, Critical Care and Sleep Medicine, Columbus, OH 43210, USA. [162]University of Texas Rio Grande Valley School of Medicine, Brownsville, TX 78520, USA. [163]Broad Institute, Harvard University, Massachusetts General Hospital, Cambridge, USA. [164]George Washington University, cardiology, Washington, DC 20037, USA. [165]University of Washington, Genome Sciences, Seattle, WA 98195, USA. [166]RTI International, North Carolina, USA. [167]Massachusetts General Hospital, Medicine, Boston, MA 02114, USA. [168]University of Arizona, Tucson, AZ 85721, USA. [169]Stanford University, Center For Sleep Sciences and Medicine, Palo Alto, CA 94304, USA. [170]National Institute of Child Health and Human Development, National Institutes of Health, Bethesda, MD 20892, USA. [171]Oklahoma Medical Research Foundation, Genes and Human Disease, Oklahoma City, OK 73104, USA. [172]Howard University, Washington, DC 20059, USA. [173]University of Maryland, Balitmore, MD 21201, USA. [174]University at Buffalo, Buffalo, NY 14260, USA. [175]University of Pennsylvania, Division of Sleep Medicine/Department of Medicine, Philadelphia, PA 19104-3403, USA. [176]Stanford University, Stanford Cardiovascular Institute, Stanford, CA 94305, USA. [177]University of Minnesota, Minneapolis, MN 55455, USA. [178]RTI International, Biostatistics and Epidemiology Division, Research Triangle Park, North Carolina 27709-2194, USA. [179]Fred Hutchinson Cancer Research Center, Fred Hutch and UW, Seattle, WA 98109, USA. [180]Johns Hopkins University, Cardiology/Medicine, Baltimore, MD 21218, USA. [181]University of Colorado at Denver, Medicine, Denver, CO 80204, USA. [182]University of Colorado at Denver, Denver, CO 80045, USA. [183]University of North Carolina, Genetics, Chapel Hill, NC 27599, USA. [184]Northwestern University, Chicago, IL 60208, USA. [185]New York Genome Center, New York Genome Center, New York City, NY 10013, USA. [186]Fred Hutchinson Cancer Research Center, University of Washington, Seattle, WA 98109, USA. [187]University of Ottawa, Sleep Research Unit, University of Ottawa Institute for Mental Health Research, Ottawa, ON K1Z 7K4, Canada. [188]Vanderbilt University, Medicine, Pharmacology, Biomedicla Informatics, Nashville, TN 37235, USA. [189]Universidade de Sao Paulo, Faculdade de Medicina, Sao Paulo 01310000, Brazil. [190]Columbia University, New York, NY 10027, USA. [191]Lundquist Institute, TGPS, Torrance, CA 90502, USA. [192]Harvard University, Division of Hematology/ Oncology, Boston, MA 02115, USA. [193]Harvard Medical School, Genetics, Boston, MA 02115, USA. [194]Harvard Medical School, Boston, MA 02115, USA. [195]Université Laval, Quebec City G1V 0A6, Canada. [196]Emory University, Pediatrics, Atlanta, GA 30307, USA. [197]Emory University, Human Genetics, Atlanta, GA 30322, USA. [198]Vanderbilt University, Medicine/Cardiology, Nashville, TN 37235, USA. [199]UMass Memorial Medical Center, Worcester, MA 01655, USA. [200]University of Saskatchewan, Saskatoon, SK S7N 5C9, USA. [201]University of Michigan, Ann Arbor, USA. [202]Albert Einstein College of Medicine, New York, NY 10461, USA. [203]Wake Forest Baptist Health, Biostatistical Sciences, Winston-Salem, NC 27157, USA. [204]University of Colorado at Denver, Genomic Cardiology, Aurora, CO 80045, USA. [205]Stanford University, Genetics, Stanford, CA 94305, USA. [206]University of Washington, University of Washington, Department of Genome Sciences, Seattle, WA 98195, USA. [207]University of Pennsylvania, Genetics, Philadelphia, PA 19104, USA. [208]University of Alabama, Biostatistics, Birmingham, AL 35487, USA. [209]University of Washington, Department of Biostatistics, Seattle, WA 98195, USA. [210]University of Southern California, USC Methylation Characterization Center, University of Southern California, California 90033, USA. [211]Brigham & Women's Hospital, Mass General Brigham, Boston, MA 02115, USA. [212]Brigham & Women's Hospital, Channing Division of Network Medicine, Department of Medicine, Boston, MA 02115, USA. [213]Massachusetts General Hospital, Boston, MA 02114, USA. [214]Indiana University, Epidemiology, Indianapolis, Indiana 46202, USA. [215]Henry Ford Health System, Detroit, MI 48202, USA. [216]Beth Israel Deaconess Medical Center, Cardiology, Cambridge, MA 02139, USA. [217]University of Pittsburgh, Medicine, Pittsburgh, PA 15260, USA. [218]University of Michigan, Department of Epidemiology, Ann Arbor, MI 48109, USA. [219]Case Western Reserve University, Department of Population and Quantitative Health Sciences, Cleveland, OH 44106, USA.

