## [Peer Review File · Nature Communications]

Whole genome sequence analysis of blood lipid levels in
>66,000 individualsREVIEWER COMMENTS

Reviewer #1 (Remarks to the Author):

In their manuscript "Whole Genome Sequence analysis of blood lipid levels in over 66,000 individuals" Selvaraj et al. describe the analyses of whole genome seq data with plasma lipid levels from the Trans-Omics for Precision Medicine (TopMed) programme. The authors investigated over 66,000 subjects across multiple ancestry groups. Results from the WGS data were analysed across and within each ancestry group and intersected with available array-based GWAS data to identify potentially novel associations of genetic variants with lipids. Replication of these results were performed in 45,000 independent samples with array-based genotyping. Subsequent analyses for identification of gene-specific functional categories and non-coding genomics regions influencing plasma lipid concentrations were performed.

Using this approach the authors identified several novel variants within known regions.

This manuscript is an interesting work using sequencing data along with microarray GWAS data to conduct a systematic scan for plasma lipids. The results show that additional alleles could be identified however, no novel loci were identified limiting the novelty of results.

My main comments is that I would like to see additional mechanistic insights of the newly identified alleles and whether these contribute to the same mechanisms as variants already know or whether novel mechanistic insights can be drawn from these variants.

Further comments:

Replication analyses of results from WGS was performed in samples with array-based genotyping imputed to TOPMed. Please explain this in more detail as one might imagine that the same results/loci/variants will be detected if our impute on the data also used for discovery.

Plasma lipid data were analysed for LDL-C, HDL-C, TC and TG. Are data on further subgroups of lipids available, eg VLDL that can be analysed (even if only available in part of the cohorts)?

Processing of sequencing data was performed in the TOPMed Informatics Research Core and the different processing steps have been described in the methods. Can the authors additionally also provide details on the rate of exclusions during processing e.g. due to sex mismatches, failure of calling etc.

Minor points:

Figure resolution needs to be improve in particular of Figure 1.

Reviewer #2 (Remarks to the Author):

In this report, Margareth Sunitha Selvaraj and colleagues performed a whole-genome sequence analysis of blood lipids in 66,329 participants from diverse ethnic groups. They replicated novel associations in up to 45,000 participants with genome-wide SNP genotyping array information. This is an original, well-designed and informative study, and the manuscript is concise and well-written.

Major comments

1-The replication group does not have whole-genome sequence information, which represents a major limitation of this study. The authors mentioned that large samples with whole-genome sequence data were scarce, but I am unsure about this claim. For instance, whole-genome sequence data have been recently made publically available for 200,000 participants from the UK Biobank. Decode Genetics has access to large samples of Icelandic participants with whole-genome sequence data.

2-As the authors have access to the information of 428 million variants, they may estimate the percentage of variation of lipid levels explained by the variants in an independent population (e.g. UK Biobank). An important question is to know if we can explain a more important fraction of heritability with WGS than SNP genotyping array data.

3-The authors may consider performing sex-specific association studies.

4-The authors may explore if lipid-associated variants show genetic pleiotropy with other complex traits (e.g. using linkage disequilibrium score regression method).

5-The authors may perform gene function / molecular pathway gene-enrichment analyses for genes located in lipid-associated loci.

6-The authors may consider prioritizing causal variants and genes at lipid-associated loci (e.g. using OpenTarget tool, Mountjoy et al., Nat Genet 2021). This is especially important to state that at least a fraction of rare non-coding variants associated with lipid levels may be causal.

7-Most of the lipid-associated loci identified by the authors using WGS have been previously identified in GWAS based on genome-wide SNP genotyping array and imputation. Therefore, it is critical to add a discussion about the pros and cons of dense WGS versus genotyping arrays in the future of genetic elucidation of complex traits. Cost is an important matter of debate here.

Minor comments

8-"Approximately 28M variants with MAC > 20 were individually associated with LDL-C, HDL-C, TC and TG". This statement is vague. Could you please add a P-value threshold?

9-"Of these variants, most were previously demonstrated to be associated with plasma lipids either at the variant- or locus-level". This statement is vague. Could you please provide a percentage?

10-The authors included non-coding variants in gene-centric rare variant analyses. They should precise which non-coding regions have been considered (introns, 5'UTR, 3'UTR) and provide information about the size of 5' and 3' regions included in the analysis

REVIEWER COMMENTS

Reviewer #1 (Remarks to the Author):

In their manuscript "Whole Genome Sequence analysis of blood lipid levels in over 66,000 individuals" Selvaraj et al. describe the analyses of whole genome seq data with plasma lipid levels from the Trans-Omics for Precision Medicine (TopMed) programme. The authors investigated over 66,000 subjects across multiple ancestry groups. Results from the WGS data were analysed across and within each ancestry group and intersected with available array-based GWAS data to identify potentially novel associations of genetic variants with lipids. Replication of these results were performed in 45,000 independent samples with array-based genotyping. Subsequent analyses for identification of gene-specific functional categories and non-coding genomics regions influencing plasma lipid concentrations were performed. Using this approach the authors identified several novel variants within known regions.

This manuscript is an interesting work using sequencing data along with microarray GWAS data to conduct a systematic scan for plasma lipids. The results show that additional alleles could be identified however, no novel loci were identified limiting the novelty of results.

1. My main comments is that I would like to see additional mechanistic insights of the newly identified alleles and whether these contribute to the same mechanisms as variants already know or whether novel mechanistic insights can be drawn from these variants.

Author Response:

Thank you for your comments and suggestions. Whether the newly identified alleles at known loci exert influence on lipids in a distinct mechanism compared to previously prioritized alleles is an interesting question and merits further investigation. In the present manuscript, we pursued extensive evaluation at the *CETP* locus. We demonstrate that African ancestry-prioritized alleles have more of an effect on LDL-C compared to HDL-C along for European ancestry-prioritized alleles. We describe genomic regions and annotations to better inform these mechanistic differences. This is of particular interest because pharmacologic *CETP* inhibition leads to starkly different LDL-C effects.

Out of the seven variants reported in Table1, three variants, "7:137875053:T:C" (*CREB3L2*), "16:56957451:C:T" (*CETP*) and "13:113841051:T:C" (*GAS6*), replicated ($p < 5 \times 10^{-5}$) in a meta-analysis of Mass General Brigham Biobank, Penn Medicine Biobank and UK Biobank. The *CETP* and *GAS6* variants were determined to be eQTLs in GTEx, prioritizing the genes that may be responsible for the variant-lipid trait associations for these two variants.

We previously showed *CETP* variants colocalized with both HDL-C and LDL-C. We have additionally performed colocalization of *GAS6* variants with eQTLs. Our *GAS6* variant, 13:113841051:T:C, is a previously reported GWAS SNP for LDL-C and TG, and our study now identified it as a novel-variant for TC. We now show that variants

associated with *GAS6* gene expression are primarily colocalized with the LDL-C and TG results, whereas for TC, we observed colocalization with *GAS6* along with several additional genes. These observations highlight the possibility of multiple cis-acting genes in addition *GAS6* which may distinguish LDL-C/TG versus TC effects.

Text in the Manuscript:

***In silico* analysis to gain mechanistic insights from single variant GWAS results Prioritization and functional enrichment analysis**

We first mapped the variants to genes and to functional regions using ANNOVAR. Second, we determined gene tissue specificity, relating tissue-specific gene expression with disease-gene associations, using MAGMA. Significantly associated variants were enriched in intronic and intergenic regions (**Supplementary Fig. 3**). Using GTEx, tissue-specific gene expression was enriched among liver, stomach, and pancreatic tissues (**Supplementary Fig. 4**) with top tissue-gene sets tabulated in **Supplementary Table 8**. Using the STRING protein-protein interaction database examining liver-specific genes, we highlight that the HDL-C protein network uniquely harbored metal-ions related genes (*MT1A*, *MT1B*, *MF1F*, *MT1G*, *MT1H*) and anticipated LCAT-CETP interactions (**Supplementary Fig. 5**). Enriched pathways from Reactome, GeneOntology and other curated and canonical pathways (**Supplementary Table 9**) with a p-value < 2.5×10^{-06} were observed including response to metal ions, lipoprotein assembly, and chylomicron remodeling.

***GAS6* locus, LDL-C, TG, and TC**

Variants at *GAS6* were previously associated with LDL-C and TG^{22 23}, but in our analysis, rs7140110 was now significantly associated with TC. We performed colocalization analysis of the variants +/-500Kb from rs7140110 in liver and adipose tissues from GTEx. Across the three lipid-related tissues (liver, adipose subcutaneous and adipose visceral), strong colocalization was observed in liver for all three lipid phenotypes (TG 46.6%; LDL-C 33.3%; TC 28%). The TG and LDL-C-associated variants were eQTLs for the *GAS6* gene only. However, the TC-associated eQTLs at this locus influenced the *cis* expression of multiple genes, including *GAS6*, antisense genes of *GAS6* (AS1, AS2) as well as other genes (i.e., *TFDP1*, *CHAMP1*, *LINC00565*, *ADPRHL1*, *RASA3*, *UPF3A*, *GRTP1*, *AL442125.1*, *C13orf46*, *DCUN1D2*, *CDC16*, *TMEM255B*, *GRTP1-AS1*, *ATP4B*, *TMCO3*). In addition to *GAS6*, the TC-associated rs7140110 is an sQTL for *TMEM255B* in adipose subcutaneous tissue (p-value 5.6×10^{-08}), with further support from TC colocalization analysis and was not significant for other lipid levels.

Methods:

Computational mining of single variant GWAS

i) Gene-set enrichment using FUMA

We performed enrichment analysis with single variant GWAS summary stats from the four lipids using FUMA⁹⁴ (version 1.3.7) with default parameters and significance at 5×10^{-9} . FUMA is an integrated platform which efficiently facilitates functional mapping and enrichment of GWAS-associated genes using multiple useful resources. The method uses 18 different biological data repositories and tools to

process GWAS data. We additionally used MAGMA⁹⁵ (version 1.08) gene-based analysis enrichment workflow within FUMA with the complete GWAS summary data for eQTL based tissue enrichment. The functionally prioritized genes were visualized based on their protein-protein interaction networks using the STRING database⁹⁶.

ii) *CETP* and *GAS6* gene expression and lipid trait colocalization

We studied the correlation of LDL-C and HDL-C effects with eQTL effects at chromosome 16q13, which includes *CETP* and correlation of LDL-C and TC with eQTLs at rs7140110 of *GAS6*. We downloaded GTEx eQTL build 38 (version8) data for liver, adipose subcutaneous, and adipose visceral (omentum) tissues from GTEx on 16/APR/2020⁹⁷. For the *CETP* variant analysis, we selected eQTLs with nominal significance ($p\text{-value} < 0.05$) and utilized the eQTL-gene pairs with the most significant p-values. Genes with at least 5 eQTLs were selected for the colocalization analysis. We selected variants with a suggestive significance ($p\text{-value} < 5 \times 10^{-7}$) for LDL-C or HDL-C effects within 500 kb of the lead locus variant. For the *GAS6* variant analysis, we curated all the GWAS variants within 500 kb of the lead variant with nominal significance ($p\text{-value} < 0.05$) and matched them to eQTL data where the transcription starting site of the corresponding gene is within ± 500 kb. We conducted colocalization analysis using the `coloc.abf()` function⁹⁸ and identified nominally significant ($PP.H4 > 1 \times 10^{-03}$) genes-eQTL pairs. The `coloc` methodology implements an efficient statistical framework to identify shared variants from two association signals through posteriors probabilities. Finally, we used the colocalized signals and compared the significant genes using STRING⁹⁶, a protein-protein interaction database. All the correlation tests were conducted in R, where we calculated Pearson correlations between the lipid effect estimates and gene expression effects (slope) from GTEx.

Further comments:

2. Replication analyses of results from WGS was performed in samples with array-based genotyping imputed to TOPMed. Please explain this in more detail as one might imagine that the same results/loci/variants will be detected if our impute on the data also used for discovery.

Author Response:

The TOPMed imputation panel is robust, built from 97,256 deeply sequenced human genomes and contains 308,107,085 genetic variants from multi-ethnic samples. This panel was used to maximize the likelihood that variants sought for replication would be present in the target dataset. Imputation procedures are agnostic to phenotypes. Most importantly, the imputation was performed in independent non-overlapping samples from Mass General Brigham Biobank, Penn Medicine Biobank, and UK Biobank.

Text in the Manuscript:

The TOPMed imputation panel is robust, built from 97,256 deeply sequenced human genomes and contains 308,107,085 genetic variants from multi-ethnic samples. Imputation was performed in independent non-overlapping samples agnostic to phenotypes.

3. Plasma lipid data were analysed for LDL-C, HDL-C, TC and TG. Are data on further subgroups of lipids available, eg VLDL that can be analysed (even if only available in part of the cohorts)?

Author Response:

LDL-C, HDL-C, TC, and TG are widely available, including in TOPMed, as they are provided by the standard lipid panel. In a prior TOPMed study⁴ we only were able to curate these measurements in a small number of individuals (n~6300) and leveraged these measures for secondary analyses after performing discovery in the standard lipid measures.

4. Processing of sequencing data was performed in the TOPMed Informatics Research Core and the different processing steps have been described in the methods. Can the authors additionally also provide details on the rate of exclusions during processing e.g. due to sex mismatches, failure of calling etc.

Author Response:

Thank you for this comment. The quality control of WGS data was carried centrally and variant discovery and genotype calling was performed jointly, across TOPMed studies, for all samples in each freeze using the GotCloud pipeline as noted by the Reviewer. Details regarding WGS data acquisition, processing and quality control vary among the TOPMed data freezes. Freeze-specific methods are described on the TOPMed website (<https://www.nhlbiwgs.org/data-sets>) along with a recent publication (Taliun D et al *Nature*. 2021). Since the central quality steps are carried out by IRC and DCC for the entire data before each freeze is released, the number of samples excluded at that stage is not considered in this study. We refer the Reviewer and readers to these external sources describing the central quality control measures of TOPMed.

Text in the Manuscript:

Quality control was performed centrally by the TOPMed IRC and the TOPMed Data Coordinating Center (DCC) as previously described¹⁷. Briefly, the two sequence quality criteria used in freeze 8 are: estimated DNA sample contamination below 10%, and 95% or more of the genome covered to 10x or greater. The variant filtering in TOPMed Freeze 8 is performed by (1) first calculating Mendelian consistency scores using known familial relatedness and duplicates, and (2) training a Support Vector Machine (SVM) classifier between known variant sites (positive labels) and Mendelian inconsistent variants. A small number of sex mismatches were detected as annotated females with low X and high Y chromosome depth or annotated males with high X and low Y chromosome depth. These samples were either excluded from the sample set to be released on dbGaP or their sample identities were resolved using information from prior array genotype comparisons and/or pedigree checks. Details regarding WGS data acquisition, processing and quality control vary among the TOPMed data freezes. Freeze-specific methods are described on the TOPMed website (<https://www.nhlbiwgs.org/data-sets>) and in documents included in each TOPMed accession released on dbGaP.

Minor points:

5. Figure resolution needs to be improve in particular of Figure 1.

Author Response:

We have increased the resolution for Figure 1.

Reviewer #2 (Remarks to the Author):

In this report, Margareth Sunitha Selvaraj and colleagues performed a whole-genome sequence analysis of blood lipids in 66,329 participants from diverse ethnic groups. They replicated novel associations in up to 45,000 participants with genome-wide SNP genotyping array information. This is an original, well-designed and informative study, and the manuscript is concise and well-written.

Major comments

1-The replication group does not have whole-genome sequence information, which represents a major limitation of this study. The authors mentioned that large samples with whole-genome sequence data were scarce, but I am unsure about this claim. For instance, whole-genome sequence data have been recently made publically available for 200,000 participants from the UK Biobank. Decode Genetics has access to large samples of Icelandic participants with whole-genome sequence data.

Author Response:

We thank the reviewer for providing this important suggestion. Whole genome sequence data in large sample sizes from diverse ancestry such as TOPMed are sparse. As suggested, we have now incorporated the UK Biobank 150K WGS dataset in this study as additional replication.

In the revised manuscript, we have provided evidence of replication for our rare variant significant results using UK Biobank whole genome sequences. For the single variant GWAS we included UKB imputed data to be consistent with other imputed datasets used for single variant GWAS replication and to utilize the large sample size of UKB imputed data. Meta-analyzed results for single variant GWAS from all the replication cohort are provided in Table 1 and Supplementary Table 5.

Importantly, using the WGS UKB data, we provide evidence of replication for all the rare variant aggregate sets from gene-centric coding, non-coding, and region-based results from sliding and dynamic windows identified in our analysis of the TOPMed data. We have tabulated the STAAR p-values the number of variants used for testing in each aggregate set in Supplementary Tables 14, 15, 16 and 17.

Text in the Manuscript:

We analyzed the UK Biobank whole genome sequences among ~130K participants to provide evidence of replication for the significant coding and non-coding aggregate sets. We used a Bonferroni-corrected significance threshold based on the number of genes tested in each type of aggregate-based test. For gene centric-coding aggregates, we conducted replication of 21 genes ($p\text{-value} < 0.05/21=2.38\times 10^{-03}$) and for non-coding aggregates we replicated the findings from 13 genes ($p\text{-value} < 0.05/13=3.85\times 10^{-03}$). At Bonferroni significance, 71% and 62% of genes replicated for at least one coding and non-coding aggregate set, respectively (**Supplementary Table 14-15**). We observed that most of the Mendelian lipid genes replicated for coding aggregates including *ABCA1*, *ABCG5*, *LCAT*, *APOB*, *LDLR*, *PCSK9*, and *LPL*. For the non-coding aggregate set, the most significant replications were observed for the *APOB*, *LDLR* (*SPC24*) and *PCSK9* loci, further corroborating the observation that both coding and noncoding rare variant signals contribute to variation in lipid levels at these loci.

We replicated 28 sliding and 51 dynamic window aggregate sets using UKB whole genomes, at a Bonferroni-corrected alpha threshold of $0.05/\text{no. of regions}$ for each approach separately. At Bonferroni significance, 61% of the regions from each of the sliding window ($p\text{-value} = 0.05/28=1.79\times 10^{-03}$) and dynamic window ($p\text{-value} = 0.05/51=9.80\times 10^{-04}$) approaches significantly replicated (**Supplementary Table 16-17**). Multiple regions linked to *LDLR*, *PCKS9*, *CETP*, *APOC3* and *ABCA1* were highly significant.

2-As the authors have access to the information of 428 million variants, they may estimate the percentage of variation of lipid levels explained by the variants in an independent population (e.g. UK Biobank). An important question is to know if we can explain a more important fraction of heritability with WGS than SNP genotyping array data.

Author Response:

We thank the reviewer for providing this suggestion. Due to our sample size for discovery with whole genome sequencing, in the revised manuscript we have included heritability estimates as implemented in Greml-LDMS approach to address the percent of variation of lipid levels that are explained using WGS data. We estimated heritability using the TOPMed WGS data from unrelated individuals in three ancestral groups (African, European, Hispanic).

We implemented the quality control approaches from a recent TOPMed paper⁵ and calculated heritability estimates by binning the variants in to 4 MAF bins. We observed an increase in heritability estimates by MAF bin. However, with the current implementation of the method, we observe that the lower MAF bins have high standard errors. Comparing our results against estimates calculated from array-genotypes⁶ from published literature and variants from MGB Biobank array-genotypes, we see a considerable increase in heritability estimates from WGS data, where the rare variants contribute to the additional proportion of variability explained. And the WGS estimates

capture the heritability from non-European ancestries more efficiently than array genotyping.

Text in the Manuscript:

Heritability contributions from rare variants:

To understand the contribution of rare variants towards lipid trait heritability, we examined heritability of lipids by variant allele frequency across three ancestral samples (White, Black, and Hispanic) in TOPMed. We calculated trait heritability using Greml-LDMS³⁹ following the steps as implemented by Wainschtein *et al*⁴⁰. Using the TOPMed WGS, we grouped the variants into 4 MAF bins for the three ancestral samples. In each MAF bin, we grouped variants based on the LD scores into 4 quartiles and calculated variance contributed by the SNPs (h^2) for each of the lipids using unrelated individuals from each ancestral group (**Supplementary Fig. 10**) and set negative estimate to zero. We observed that rare variants from the lower MAF bins contributed to trait heritability but have large standard errors (**Supplementary Table 20**). We observed an increase in h^2 values including WGS variants relative to estimates obtained from array-genotypes as reported by Cadby *et al*⁴¹ for the European samples. We also compared the h^2 estimates from all the variants from WGS TOPMed cohort against array-genotypes captured in MGB Biobank to understand the differences contributed by these two sequencing methods. As expected, the h^2 estimates from array-genotypes were reduced corresponding to missing heritability from the lower MAF bins captured by WGS. The heritability estimates from array-genotypes were markedly higher for European samples relative to African and Hispanic sample sets indicating that WGS better captured heritability for the latter groups.

Methods:

Calculation of heritability estimates from TOPMed WGS data

We calculated heritabilities estimated for the four lipids using TOPMed WGS data using Greml-LDMS approach³⁹, where we binned the variants into four MAF bins based on minor allele frequency and grouped the variants to four LD quartiles based on LD score calculated by GCTA method⁹⁹. The four MAF bins used in this study includes ≥ 0.05 , ≥ 0.01 to < 0.05 , ≥ 0.001 to < 0.01 and ≥ 0.0001 to < 0.001 . We excluded any variant with $MAF < 0.0001$ from this analysis. The hereditary estimation was calculated for three ancestral groups (African, European, Hispanic) where only unrelated samples (kinship score < 0.025) were included in the analysis. We excluded the other two ancestral groups (i.e., Asian and Samoan) from this analysis due to insufficient sample sizes. In total we included 9640, 21568 and 10631 in African, European and Hispanic ancestries respectively. For each MAF bin, we implemented certain quality control (QC) measures using PLINK software²⁰, which includes; genotype missingness (--geno 0.05), sample missingness (--mind 0.05), Hardy-Weinberg equilibrium (--hwe 10^{-6}) and LD pruned variants (--indep-pairwise 50 5 0.1) as implemented by Wainschtein *et al*⁴⁰. Next, we implemented Greml-LDMS with LD score region as 200 and GRM cut-off as 0.05 for the four lipid phenotypes. We calculated 20 principal components from the QC passed variants in each MAF bin and implemented GCTA workflow with --reml-no-

constrain, --reml-no-lrt and --reml-maxit 10000 parameters to avoid the no-convergence issues and negative h^2 estimates. For comparing the h^2 estimates between variants from WGS data and array-genotypes, first, we used QC passed WGS variants as mentioned above, second, we curated the variants from MGB Biobank array data and intersected them with WGS variants from TOPMed. Next, we calculated heritability estimates for array-genotype variants and compared with h^2 estimates from WGS variants for the three ancestral groups.

3-The authors may consider performing sex-specific association studies.

Author Response:

We thank the reviewer for this comment. We would like to refer the reviewer to a recent Global Lipids Genetics Consortium manuscript⁷ that reports sex-specific associations for lipid levels in a much larger analysis using array-based genotypes. In that study, we performed a GWAS meta-analysis separately in males (N=749,391) and females (N=562,410) and excluded loci discovered in the sex-combined analysis. We identified few loci that reached genome-wide significance only in one sex ($p < 5 \times 10^{-8}$; 16 in females and 9 in males). Given the low yield of our analyses in a sex-combined analysis and the much more robust finding from the GWAS, we believe we would be relatively underpowered to detect additional effects using the TOPMed WGS data.

4-The authors may explore if lipid-associated variants show genetic pleiotropy with other complex traits (e.g. using linkage disequilibrium score regression method).

Author Response:

To understand the genetic correlation with other complex traits, we conducted a phenome-wide association (PheWAS) using 1572 complex traits in the UK Biobank. We extended this analysis for three common variants which showed significant replication, which are 16:56957451:C:T (*CETP*); 13:113841051:T:C (*GAS6*); 7:137875053:T:C (*CREB3L2*). We identified complex traits which were significant at $FDR < 0.05$ and tabulated the results in Supplementary table 11.

Text in the Manuscript:

Phenome wide association with complex traits:

We conducted a phenome-wide association (PheWAS) of 1572 binary complex traits using UK Biobank for the three replicated common variants (16:56957451:C:T (*CETP*); 13:113841051:T:C (*GAS6*); 7:137875053:T:C (*CREB3L2*)). We adjusted for PC1-10, age, age², sex and race, for each trait. We claimed significance at $FDR 0.05$ and identified various complex traits significant, including ischemic heart disease for the *CETP* variant and heart failure/atherosclerosis, hypercholesterolemia traits for *GAS6* variant. The summary statistics from PheWAS analysis for the significant complex traits are tabulated in **Supplementary Table 11**.

Methods:

iii) Phenome wide association analysis

The complex trait information was curated from UK Biobank resource, where we curated multiple disease phenotypes for UKB samples into International Classification of Diseases (ICD)-based phecodes based on phecode map (<https://phewascatalog.org>) using the PheWAS R package (version PheWAS_0.99.5-4). We conducted a phenome-wide association analysis (PheWAS) using a logistic regression model `glm()` in R. We adjusted the models for PC1-10, age, age², sex, and race.

5-The authors may perform gene function / molecular pathway gene-enrichment analyses for genes located in lipid-associated loci.

Author Response:

Please refer to R #1 Point #1 above.

6-The authors may consider prioritizing causal variants and genes at lipid-associated loci (e.g. using OpenTarget tool, Mountjoy et al., Nat Genet 2021). This is especially important to state that at least a fraction of rare non-coding variants associated with lipid levels may be causal.

Author Response:

Open targets and specifically the Open Targets Genetics Portal integrates public domain GWAS data to prioritize variants and targets. But most of the integrated data comes from common variant studies and a very few rare variants are integrated to the portal. Therefore, we queried the single variant GWAS SNPs using the tool but did not identify them as causal variants. We carried out variant prioritization and the results are described above.

Text in the Manuscript:

We did not find information for these variants in the Open Target Genetics database²⁸. Finally, two of the common novel-loci variants (rs183130 and rs7140110) were present in eQTL and sQTL databases²⁹, and we performed analysis to determine the correlation among effects for these variants more in detail.

7-Most of the lipid-associated loci identified by the authors using WGS have been previously identified in GWAS based on genome-wide SNP genotyping array and imputation. Therefore, it is critical to add a discussion about the pros and cons of dense WGS versus genotyping arrays in the future of genetic elucidation of complex traits. Cost is an important matter of debate here.

Author Response:

We thank the reviewer for this comment. We have added the comparison in the Discussion.

Text in the Manuscript:

Our discovery analyses with replication as well as heritability assessment are consistent with the notion that both rare coding and non-coding alleles, not well-captured by genome-wide arrays. Furthermore, we observe that heritability gains relative to

genome-wide genotyping arrays are more significant for individuals of European-ancestry likely indicative of Eurocentric array designs. A tradeoff for WGS, however, is the greater cost. However, as costs continue to decrease as well as cheaper WGS implementations via reduced coverage, cost may no longer be a downside.

Minor comments

8-“Approximately 28M variants with MAC > 20 were individually associated with LDL-C, HDL-C, TC and TG”. This statement is vague. Could you please add a P-value threshold?

Author Response:

To clarify the sentence, we have modified the text. There is no p-value thresholding at this stage, we used a MAC cut-off of 20 to start the single variant GWAS. Therefore, in the discovery cohort, we started with approximately 28M variants.

Text in the Manuscript:

We performed single variant analysis of approximately 28M variants with a MAC > 20 for four lipid phenotypes. We identified significant genomic risk loci for each lipid level (**Supplementary Table 3**) and considered a p-value < 5×10^{-9} to claim significance as previously recommended for whole genome sequencing common variant association studies.

9-“Of these variants, most were previously demonstrated to be associated with plasma lipids either at the variant- or locus-level”. This statement is vague. Could you please provide a percentage?

Author Response:

We now provide a percentage based on our discovery data in the sentence.

Text in the Manuscript:

Of these variants, 99% were previously demonstrated to be associated with plasma lipids either at the variant- or locus-level.

10-The authors included non-coding variants in gene-centric rare variant analyses. They should precise which non-coding regions have been considered (introns, 5'UTR, 3'UTR) and provide information about the size of 5' and 3' regions included in the analysis

Author Response:

We included 7 non-coding mask in our analysis based on STAAR workflow⁸, as described in the text. The UTR mask includes rare variants in both 5' and 3' UTR regions and the UTR region is defined by GENCODE Variant Effect Predictor (VEP) categories. We have added more details about these masks and the defining criteria.

Text in the Manuscript:

We aggregated rare variants into multiple groups for coding and non-coding analyses. For the coding region, we defined five different aggregate masks of rare variants 1) plof (putative loss-of-function), plof-Ds (putative loss-of-function or disruptive missense), missense, disruptive-missense, and synonymous. For the non-coding regions, we used seven rare variant masks: 1) promoter-CAGE (promoter variants within Cap Analysis of Gene Expression [CAGE] sites^{86,87}), 2) promoter-DHS (promoter variants within DNase hypersensitivity [DHS] sites⁸⁸), 3) enhancer-CAGE (enhancer within CAGE sites⁸⁷), 4) enhancer-DHS (enhancer variants within DHS sites⁸⁹), 5) UTR (rare variants in 3' untranslated region [UTR] and 5' UTR untranslated region), 6) upstream, and 7) downstream. Detailed explanations of the regions defined based on these masks is discussed within STAARpipeline¹³.

In the gene-centric workflows, for both coding (within exonic boundaries) and non-coding (promoter: +/- 3kb window of transcription starting site (TSS), enhancer: GeneHancer predicted regions, UTR (both 5' and 3' UTR regions)/upstream/downstream: GENCODE Variant Effect Predictor (VEP) categories) regions, we considered only genes with at least two rare variants (i.e., 18,445 genes in all 22 autosomes).

References:

1. Subramanian, A. *et al.* Gene set enrichment analysis: A knowledge-based approach for interpreting genome-wide expression profiles. *Proc. Natl. Acad. Sci.* **102**, 15545–15550 (2005).
2. Mountjoy, E. *et al.* An open approach to systematically prioritize causal variants and genes at all published human GWAS trait-associated loci. *Nat. Genet.* **53**, 1527–1533 (2021).
3. Lonsdale, J. *et al.* The Genotype-Tissue Expression (GTEx) project. *Nat. Genet.* **45**, 580–585 (2013).
4. Natarajan, P. *et al.* Chromosome Xq23 is associated with lower atherogenic lipid concentrations and favorable cardiometabolic indices. *Nat. Commun.* **12**, 2182 (2021).
5. Wainschein, P. *et al.* Assessing the contribution of rare variants to complex trait heritability from whole-genome sequence data. *Nat. Genet.* **54**, 263–273 (2022).
6. Cadby, G. *et al.* Heritability of 596 lipid species and genetic correlation with cardiovascular traits in the Busselton Family Heart Study. *J. Lipid Res.* **61**, 537–545 (2020).
7. Kanoni, S. *et al.* *Implicating genes, pleiotropy and sexual dimorphism at blood lipid loci through multi-ancestry meta-analysis.*
<http://medrxiv.org/lookup/doi/10.1101/2021.12.15.21267852> (2021)
doi:10.1101/2021.12.15.21267852.
8. Li, X. *et al.* Dynamic incorporation of multiple in silico functional annotations empowers rare variant association analysis of large whole-genome sequencing studies at scale. *Nat. Genet.* **52**, 969–983 (2020).

REVIEWERS' COMMENTS

Reviewer #1 (Remarks to the Author):

The authors provide a very good reply to my comments, and can be congratulated on this nice work. All my comments were revised sufficiently.

Reviewer #2 (Remarks to the Author):

The authors addressed adequately all my comments, thank you.

REVIEWERS' COMMENTS

Reviewer #1 (Remarks to the Author):

The authors provide a very good reply to my comments, and can be congratulated on this nice work.

All my comments were revised sufficiently.

Authors response:

We thank the reviewer for the positive comment.

Reviewer #2 (Remarks to the Author):

The authors addressed adequately all my comments, thank you.

Authors response:

We thank the reviewer for the positive comment.